# Heterogeneous ice nucleation on dust particles sourced from 9 deserts worldwide - Part 1: Immersion freezing

Yvonne Boose[1], André Welti[1,2], James Atkinson[1], Fabiola Ramelli[1], Anja Danielczok[3], Heinz G. Bingemer[3], Michael Plötze[4], Berko Sierau[1], Zamin A. Kanji[1], and Ulrike Lohmann[1]

[1]Institute for Atmospheric and Climate Science, ETH Zürich, Zürich, Switzerland
[2]now at Leibniz Institute for Tropospheric Research, Leipzig, Germany
[3]Institute for Atmospheric and Environmental Sciences, J. W. Goethe-University, Frankfurt am Main, Germany
[4]Institute for Geotechnical Engineering, ETH Zürich, Zürich, Switzerland

*Correspondence to:* Y.Boose (yvonne.boose@env.ethz.ch) and Z. A. Kanji (zamin.kanji@env.ethz.ch)

**Abstract.** Desert dust is one of the most abundant ice nucleating particle types in the atmosphere. Traditionally, clay minerals were assumed to determine the ice nucleation ability of desert dust and constituted the focus of ice nucleation studies over several decades. Recently some feldspar species were identified to be ice-active at much higher temperatures than clay minerals, redirecting studies to investigate the contribution of feldspar to ice nucleation on desert dust. However, so far no study has shown the atmospheric relevance of this mineral phase.

For this study four dust samples were collected after airborne transport in the troposphere from the Sahara to different locations (Crete, the Peloponnese, Canary Islands and the Sinai Peninsula). Additionally, eleven dust samples were collected from the surface from nine of the biggest deserts worldwide. The samples were used to study the ice nucleation behavior specific to different desert dusts. Furthermore we investigated how representative surface-collected dust is for the atmosphere by comparing to the ice nucleation activity of the airborne samples. We used the IMCA-ZINC set-up to form droplets on single aerosol particles which were subsequently exposed to temperatures between 233 - 250 K. Dust particles were collected in parallel on filters for offline cold stage ice nucleation experiments at 253 - 263 K. To help the interpretation of the ice nucleation experiments the mineralogical composition of the dusts was investigated. We find that a higher ice nucleation activity in a given sample at 253 K can be attributed to the K-feldspar content present in this sample whereas at temperatures between 238 - 245 K it is attributed to the sum of feldspar and quartz content present. A high clay content on the other hand is associated with lower ice nucleation activity. This confirms the importance of feldspar above 250 K and the role of quartz and feldspars determining the ice nucleation activities at lower temperatures as found by earlier studies for monomineral dusts. The airborne samples show on average a lower ice nucleation activity than the surface-collected ones. Furthermore, we find that under certain conditions milling can lead to a decrease in the ice nucleation ability of polymineral samples, due to the different hardness and cleavage of individual mineral phases causing an increase of minerals with low ice nucleation ability in the atmospherically relevant size fraction. Comparison of our data set to an existing desert dust parameterization confirms its applicability for climate models. Our results suggest that for an improved prediction of the ice nucleation ability of desert dust in the atmosphere, the modelling of emission and atmospheric transport of the feldspar and quartz mineral phases would be key while other minerals are only of minor importance.

# 1 Introduction

Predicting the occurrence and evolution of clouds at temperatures ($T$) below 273 K remains a challenge for global and regional climate models (Boucher et al., 2013). One source of uncertainty is the effect of certain aerosol particles which influence the cold cloud microphysics by acting as ice nucleating particles (INPs). Ice formation affects precipitation, cloud life time and radiative properties of these clouds and by that global climate (Lohmann and Feichter, 2005). Mineral dust particles have been known as efficient INPs at $T \leq 253$ K for more than 60 years (e.g. Isono 1955, and references given in Hoose and Möhler 2012; Murray et al. 2012) and have been observed to nucleate ice in the atmosphere in various regions worldwide (Kumai, 1976; DeMott et al., 2003; Chou et al., 2011; Boose et al., 2016a, b). However, the molecular mechanisms and particle properties triggering ice nucleation on atmospheric mineral dusts are still subject of ongoing research. Supercooled cloud droplets can freeze homogeneously at temperatures below 235 K, without the aid of an INP (Schaefer, 1946; Mason and Ludlam, 1950). At higher temperatures the surface of an INP is required to overcome the energy barrier of freezing. Traditionally, four pathways of ice nucleation are differentiated (Vali et al., 2015):

1) deposition nucleation where ice forms on an INP directly from the vapor phase;

2) condensation freezing, in which ice forms during the process of water condensing on an INP;

3) immersion freezing where an INP immersed in a supercooled cloud droplet initiates freezing;

4) contact freezing where the interaction of an INP with the surface of a supercooled droplet either from the outside or inside of the droplet leads to freezing.

Ice formation in clouds with top temperatures above 263 K is often observed (Hobbs and Rangno, 1985) but only very few aerosol particle types have been identified to nucleate ice at these warm temperatures. These are mainly biological particles, such as certain bacterial strains or macromolecules (Schnell and Vali, 1976; Krog et al., 1979; Möhler et al., 2008b; Pummer et al., 2012). The ice nucleation ability of soot (Brooks et al., 2014; Kulkarni et al., 2016) at heterogeneous freezing temperatures is still debated as contradicting results were observed, spanning from hardly any ice nucleation ability at $T > 236$ K (Kanji et al., 2011) to up to 3 % of soot particles active in the immersion mode (DeMott, 1990). Similarly, the reported freezing behavior of secondary organic aerosol particles varies from inefficient to comparably efficient (Möhler et al., 2008a; Prenni et al., 2009; Wang et al., 2012; Ladino et al., 2014; Ignatius et al., 2016). Aerosol particles from marine sources are believed to be important INP at remote locations and are subject of current research (Knopf et al., 2011, 2014; DeMott et al., 2015a; Wilson et al., 2015). Recently, the K-feldspar microcline and the Na-feldspar albite, both minerals found in atmospheric dust, have been identified to nucleate ice at temperatures up to 271 K (Harrison et al., 2016).

For the implementation of ice nucleation into climate models, a simplistic description of ice formation on different INP types is required. Existing parameterizations for dust can be based on laboratory experiments using commercially available dusts such as Arizona Test Dust (ATD) including mostly pure clay mineral samples such as illite, kaolinite or montmorillonite (Lüönd et al., 2010; Murray et al., 2011; Niedermeier et al., 2011), dust samples collected from the surface (Niemand et al., 2012) or on in-situ measurements in the atmosphere at locations often distant from major dust sources (DeMott et al., 2010; Tobo et al., 2013). One recent study by DeMott et al. (2015b) combines laboratory data of two surface-collected dust samples with

results from two flight campaigns over the Pacific Ocean and the Caribbean Sea within dust layers that underwent long-range transport from Asia and the Sahara, respectively. The authors found relatively good agreement amongst the different samples. They concluded that both a parameterization from Niemand et al. (2012) as well as one adapted from Tobo et al. (2013) were applicable for predicting atmospheric mineral dust INP concentrations.

For laboratory ice nucleation experiments, dust samples collected from the surface typically have to be sieved or milled, which may break up larger agglomerates and alter the size-dependent mineralogy (Perlwitz et al., 2015). This could significantly alter the ice nucleation ability of these dust particles in laboratory experiments compared to their ambient ice nucleation ability. It has been shown that milling of hematite or quartz particles leads to an increase in ice nucleation efficiency compared to the unmilled samples (Hiranuma et al., 2014; Zolles et al., 2015). It has been speculated that this is also part of the reason for ATD,

a commercially available dust sample that is washed and milled after collection from a certain desert area in Arizona, being more ice nucleation-active than natural unprocessed dust samples (Möhler et al., 2006).

Due to their high abundance, for many decades the immersion freezing behavior of atmospheric dust was attributed largely to clay minerals and ice nucleation on relatively pure clay mineral samples were often studied in more detail (Hoffer, 1961; Lüönd et al., 2010; Murray et al., 2010, 2011; Broadley et al., 2012; Pinti et al., 2012; Welti et al., 2012; Hiranuma et al., 2015).

Recently, Atkinson et al. (2013) showed that compared to other minerals feldspar particles are more efficient immersion mode INP at temperatures above 245 K. The K-feldspars (microcline, orthoclase and sanidine) were found to be more ice nucleation-active than the Na/Ca-feldspars albite, anorthite and other plagioclase feldspars (Atkinson et al., 2013; Zolles et al., 2015; Peckhaus et al., 2016). Amongst the K-feldspars microcline appears to be the most ice nucleation-active (Augustin-Bauditz et al., 2014; Kaufmann et al., 2016), even nucleating ice at a temperature of 271 K (Harrison et al., 2016). Feldspar is a highly

complex group of minerals and, depending on the source, mineralogically similar samples can have different ice nucleation abilities (Harrison et al., 2016). Thus it remains an open question if and how feldspar is affecting the ice nucleation behavior of dust in the atmosphere and if it is causing ice nucleation in clouds at $T > 263$ K. A high variability in ice nucleation activity was found for quartz, with some quartz samples being more ice nucleation-active in the immersion mode than clay minerals but always less than the feldspars (Atkinson et al., 2013; Zolles et al., 2015; Kaufmann et al., 2016). It is suspected

that functional groups on the surface of feldspars and quartz are responsible for their higher ice nucleation ability (Zolles et al., 2015) but it is unknown where the high variability stems from. Quartz is commonly (5-50 wt%) found in atmospherically transported Saharan dust samples (Avila et al., 1997; Caquineau et al., 1998; Alastuey et al., 2005; Kandler et al., 2009). A recent study by Kaufmann et al. (2016) investigated the ice nucleation ability of surface-collected samples from eight different arid regions worldwide and several single-mineral reference samples using differential scanning calorimetry. The authors found at

maximum a 6 K spread in freezing temperatures of emulsion experiments amongst surface-collected samples from different atmospheric-dust source regions. They confirmed the exceptional freezing ability of microcline but found only a minor fraction (4 wt%) in one of the samples from the dust source regions studied. Their samples contained quartz fractions between 1 and 26 wt%, K-feldspar fractions between 0 and 10 wt%, and plagioclase fractions between 0 and 22 wt%.

It has been observed that the size distribution of dust changes during its emission and transport compared to dust on the sur-

face. This leads to variations in the mineralogical composition of the dust (D'Almeida and Schütz, 1983; Murray et al., 2012;

Knippertz and Stuut, 2014), as the mineralogical composition is size dependent due to differences in the hardness, cleavage, shape and reactivity of minerals. Hard minerals such as feldspar tend to be dominant in the large grains whereas soft minerals are concentrated in the small size fraction (e.g. clay minerals). Saltation and dust emission strength depends on several factors and is nonlinear in dust particle size (Knippertz and Stuut, 2014). During atmospheric transport, gravitatonal settling or wet deposition further alters the size distribution. Additionally, minerals which act as cloud condensation nuclei or INP are preferably lost.

Airborne dust particles smaller than 20 $\mu$m over the North Pacific have been found to contain 10 to over 50 wt% clay minerals such as illite, kaolinite or smectite, 4 - 40 wt% quartz and 4 - 75 wt% plagioclase feldspar (Leinen et al., 1994). Kandler et al. (2009) found that dust particles over Morocco consist of about 30 wt% clay minerals (illite, kaolinite, chlorite), less than 5 wt% plagioclase but over 20 wt% K-feldspar, less than 10 wt% quartz and less than 10 wt% calcite in the size range below about 20 $\mu$m geometric diameter. Other identified minerals in the airborne dust were rutile, gypsum, dolomite, hematite or halite. Similar results were found by Falkovich et al. (2001) over Israel. Caquineau et al. (1998) found a north-south gradient of the illite to kaolinite ratio of soil samples in the Sahara with higher values in the northern and western part of the Sahara and lower values in the southern and central Sahara.

Non-mineral matter, which can become internally or externally mixed with the mineral dust before or after emission, may affect the ice nucleating behavior of the dust. Sulfuric acid (Sullivan et al., 2010; Augustin-Bauditz et al., 2014) or secondary organic aerosol coating (Möhler et al., 2008a) has been observed to decrease the ice nucleating ability while exposure to ozone (Kanji et al., 2013) or the presence of ammonium sulfate (Boose et al., 2016b) has been suggested to improve it. Biological material can adsorb to mineral dust, enhancing its ice nucleating ability (Schnell, 1977; Conen et al., 2011; O'Sullivan et al., 2016).

In this study we investigate the immersion ice nucleation properties of 15 dust samples from nine different deserts around the world. Four of the samples were collected directly from the air (Tenerife) or by deposition after atmospheric transport (Crete, Egypt, Peloponnese) for subsequent analysis in the laboratory without additional treatment such as sieving or milling. Based on back trajectory analysis, the four airborne samples originate from different parts of the Sahara. The ice nucleation ability of these airborne dusts was compared to that of several samples collected in the desert. The effect of sieving and milling on the ice nucleation behavior of two surface-collected samples was investigated.

Immersion mode ice nucleation measurements at temperatures between 235 and 250 K were conducted with the combination of the Zurich Ice Nucleation Chamber, ZINC (Stetzer et al., 2008), and the Immersion Mode Cooling chAmber, IMCA (Lüönd et al., 2010). Particles of four dust samples were collected on filters for subsequent offline analysis with the Frankfurt Ice Deposition Freezing Experiment (FRIDGE) counter operated in the droplet freezing mode as described by Ardon-Dryer and Levin (2014) and Hiranuma et al. (2015). This allowed examination of immersion freezing at temperatures between 250 and 262 K, covering a wider range of heterogeneous freezing temperatures than would otherwise be possible with IMCA-ZINC alone. The aim of the current, and a follow-up study on deposition/condensation nucleation, is to investigate the link between ice nucleation and bulk mineralogy of desert dust as it is found in the atmosphere and to compare it to surface-collected samples. By using aeolian transported samples, the particle size distribution and sample composition are as realistic as possible. To our

knowledge this is the first study to investigate ice nucleation behavior of airborne desert dust in the laboratory, compare it with surface-collected natural dust samples and link it to the mineralogical composition of these complex samples. With samples from nine different deserts we present a data set covering most major global dust sources.

## 2 Methods

### 2.1 Dust sample origins and processing

The immersion mode freezing behavior of a total of 15 different dust samples was investigated. The collection sites are shown in Fig. 1 together with the major dust emission sources and common atmospheric transport pathways. GPS coordinates of the collection sites are provided in the supplementary material. It can be seen from Fig. 1 that the dust samples stem from most of the major atmospheric dust sources. The Tenerife sample was collected directly from the air over four days in August 2013 at the Izaña observatory on Tenerife, Spain, using a custom-made large cyclone (Advanced Cyclone Systems, S.A.: flow rate: 200 m$^3$h$^{-1}$, $D_{50}$ = 1.3 $\mu$m, the diameter at which the collection efficiency is 50 %). After deposition on a roof and on solar panels, dust samples were collected at the Aburdees observatory, Egypt on 10 May, 2010 and in Crete and the Peloponnese in Greece in April, 2014. The Crete sample was an integrated sample over several dust events whereas the Peloponnese sample was from one single dust event. Surface collection sites were (i) the Atacama desert in Chile; (ii) a location approximately 70 km from Uluru in Australia; (iii) the Great Basin in Nevada and (iv) the Mojave desert in California, USA; (v) a Wadi in the Negev desert, approximately 5 km from Sde Boker in Israel; (vi) dunes in the Sahara, close to Merzouga in Morocco; (vii) dunes in the Arabian desert in Dubai; (viii) the Etosha pan in Namibia, a dry salt pan; and (ix) the Taklamakan desert in China. The Israel sample and the Etosha sample are from the same batch as those studied in Kaufmann et al. (2016).

The surface-collected samples needed to be sieved to separate the grain sizes larger than 32 $\mu$m from the remaining sample to avoid clogging of the aerosol generation system used for the ice nucleation experiments. Samples were sieved in a cascade of dry sieves with the smallest cut-off size being at 32 $\mu$m diameter (Retsch Vibratory Sieve Shaker AS 200). Typically only a few weight percent of the sample was in this size range. The Australia and Morocco samples had no fraction in this size range and thus were milled using a vibratory disc mill (Retsch, model RS1). For the Morocco sample, particles of the lowest size bin (32 to 64 $\mu$m) were milled. The Australia sample was first sieved with a coarse, millimeter range sieve to separate any large material, and the remaining sand was milled. For the Atacama and Israel samples, both a milled and a sieved sample were compared to investigate the effect of milling on ice nucleation. In case of the Atacama sample, part of the unsieved sample was milled. The Israel sample was first sieved and part of the sieved sample with $d \leq 32$ $\mu$m was milled. The sub-32 $\mu$m size fraction of the other samples was too small to investigate the milling effect. The composition of natural dust samples is presumed to be heterogeneous, i.e. external and internal mixtures of different minerals and potentially containing organic or biological material (Meola et al., 2015). Additionally, they have probably undergone natural aging processes due to the exposure to the atmosphere of the surface-collected samples and actual atmospheric aging of the airborne samples (Dall'Osto et al., 2010). This could physically or chemically alter the surface of the dust particles, potentially changing the ice nucleation properties compared to the pure mineral dust particles. Effects of washing or heating of the samples, which could yield information on

coating or mixing, could not be investigated in this part of the study due to the small sample size of the airborne samples.

## 2.2 Dust particle generation

The dust samples were dry dispersed into a 2.78 m$^3$ stainless steel aerosol reservoir tank (Kanji et al., 2013) using a Rotating

Brush Generator (RBG, Palas, model RBG 1000) with N$_2$ (5.0) as carrier gas via a cyclone that confined the dust size distribution to below $D_{50}$ = 2.5 $\mu$m. The maximum particle concentration in the tank was about 1200 cm$^{-3}$ and decreased steadily to about 300 cm$^{-3}$ over approximately 10 h. Before each experiment, the tank was cleaned by repeatedly evacuating and purging it with N$_2$ until the particle concentration decreased to 30 - 90 cm$^{-3}$. The total particle concentration was monitored with a Condensation Particle Counter (CPC; TSI model 3772). The ice nucleating particle counters, the particle collection for offline

FRIDGE experiments and the instruments measuring the particles' size distribution sampled directly from the tank. For the IMCA-ZINC measurements the particle concentration was diluted to about 60 cm$^{-3}$ to avoid coincidence effects in the detector which occur if more than one particle is present in the laser beam of the detector (Nicolet et al., 2010).

## 2.3 Aerosol particle size distribution

The particle size distribution in the reservoir tank was monitored using a Scanning Mobility Particle Sizer (SMPS; TSI; DMA model 3081, CPC model 3010) for mobility diameters ($d_{\mathrm{m}}$) between 12.2 - 615 nm and an Aerodynamic Particle Sizer (APS; TSI; model 3321) for aerodynamic diameters ($d_{\mathrm{aer}}$) between 0.5 - 20 $\mu$m. After converting the mobility and aerodynamic diameter to volume equivalent diameter ($d_{\mathrm{ve}}$) the size distributions were merged. A shape factor of $\chi$ = 1.36 and a particle density of $\rho$ = 2.65 g cm$^{-3}$ were assumed for the conversion. These values lie in the range of natural dust samples analyzed in

earlier studies, e.g. quartz: $\chi$ = 1.10-1.36 (Hinds, 1999; Alexander, 2015), $\rho$ = 2.6 g cm$^{-3}$ (Hinds, 1999; Kandler et al., 2007), illite NX: $\chi$ = 1.49 and $\rho$ = 2.65 g cm$^{-3}$ (Hiranuma et al., 2015). Assuming spherical particles, the area size distribution was calculated and fitted with a bimodal lognormal distribution. The mean particle surface area ($\overline{A_{\mathrm{ve,w}}}$) was calculated from the resulting fit for each sample (see Table 1) as well as the corresponding surface area-weighted mean diameter ($\overline{d_{\mathrm{ve,w}}}$). Over the course of a single experiment the size distribution changed as larger particles settle out of the volume faster than smaller ones.

This effect was reduced by a fan inside the aerosol tank leading to $\overline{A_{\mathrm{ve,w}}}$ varying by 6 to 24 % over the course of an experiment for the different samples, except for the Great Basin sample (64 %), which was coarser than the other samples and settled out faster.

Figure 2 shows a schematic of the different size fractions resulting from the different collection methods and post-treatment (sieving/milling) of the samples used for ice nucleation experiments in the tank and the mineralogical analysis. Due to the small

amount of sample, a mineralogical analysis of the identical size fraction as in the tank ($<$ 2.5 $\mu$m) was not possible. Instead, we used the entire size fraction of the airborne samples, the smallest size fraction of the sieved samples ($<$ 32 $\mu$m), and of the milled samples the whole size distribution after milling.

## 2.4 Mineralogy analysis

The quantitative mineralogical composition of the bulk dust samples was investigated using the X-ray diffraction (XRD) Rietveld method (Rietveld, 1969) using a Bragg-Brentano diffractometer (Bruker AXS D8 Advance with CoKalpha-radiation). The qualitative phase composition was determined with the software DIFFRACplus (Bruker AXS). On the basis of the peak positions and their relative intensities, the mineral phases were identified in comparison to the PDF-2 data base (International Centre for Diffraction Data). The quantitative composition was calculated by means of Rietveld analysis of the XRD pattern (Rietveld program AutoQuan, GE SEIFERT, Bergmann et al. 1998; Bish and Plötze 2011).

The results and uncertainties for the mineralogy of each sample given from the Rietveld refinement are provided in Tables 2 and 3. For the Egypt sample the mineralogical composition is associated with a significantly higher uncertainty because the amount of sample was small and the measured intensity of the diffracted X-rays very low. In this case the grain statistics were poor and crystals more likely to be arranged in a certain, preferred orientation instead of randomly, leading to a potential overestimation of some mineral fractions. Similarly, the milling of the Israel sample likely interfered with the preferred orientation of the minor components in the sieved samples, leading to an observed reduction of these mineral fractions (e.g. illite, dolomite, plagioclase) in the milled compared to the sieved sample. The differentiation between the microcline and orthoclase K-feldspar fraction was for some samples not possible (i.e. Morocco and Australia) where both phases were likely present. In case of a low K-feldspar content of a few wt% it was not possible to determine if K-feldspar was present as microcline, orthoclase or sanidine or a mixture of the different phases. Values are given for the K-feldspar with the best Rietveld fit result. As Rietveld fit results for the various Na-plagioclase feldspars (albite, oligoclase and andesine) were often insignificantly different, they are summarized as Na-plagioclase. The fraction of ankerite and dolomite are usually provided together because in some cases (especially Morocco) it was not possible to differentiate between them. Only for the Etosha sample they are provided separately because the fractions were large enough to be distinguishable (Kaufmann et al., 2016).

Due to the broader size range of particles studied with XRD, the mineralogy is not only describing the particles that were studied with the ice nucleation chambers but also the fraction between 2.5 - 32 $\mu$m. This could lead to differences between the measured mineralogy and the actual mineralogical composition of particles smaller than 2.5 $\mu$m due to differences in the hardness and cleavage or fracture, i.e. the breaking behavior, of different minerals. This is particularly true for softer minerals such as calcite, which has a Mohs hardness of 3 (standard scale of hardness between 1, talc and 10, diamond), and clay minerals (2 - 2.5) in contrast to feldspars (6) and quartz (7) as well as for minerals with a higher cleavage such as gypsum and calcite (perfect cleavage) compared to quartz (without cleavage, for information on mineral cleavage and hardness see www.mindat.org or www.webmineral.com). Natural mechanical weathering thus likely has enhanced the clay mineral and calcite content in the smaller particle fraction whereas feldspars and quartz tend to be found in the larger size fractions. For each filling of the reservoir tank a similar volume of dust sample was used ($\approx 0.2$ cm$^3$) and the dust density is assumed to be comparable (about 2.65 g cm$^{-3}$). This allows to roughly approximate what fraction of particles was larger than 2.5 $\mu$m by the amount of dust sample left over in the 2.5 $\mu$m cut-off cyclone and the particle concentration reached in the tank. Hardly any particles were left over in the cyclone and maximum particle concentrations of 900 - 1200 cm$^{-3}$ were reached by all milled

samples apart from the Morocco sample, and by all airborne samples apart from the Egypt sample. We suspect that the Egypt sample has a higher fraction of large particles because it originated from local sources within Egypt and thus the transport time was much shorter compared to the other airborne samples leading to the size distribution being shifted to larger particles. Of the sieved samples, the Dubai, Great Basin, Israel, Mojave and Taklamakan samples had a comparably high fraction of particles larger than 2.5 $\mu$m and particle concentrations of 400 - 970 cm$^{-3}$ were reached when filling the tank. For these samples the presented mineralogy may not be fully representative for the particles $< 2.5$ $\mu$m investigated for ice nucleation. In contrast, particles of the Etosha and Atacama sieved samples were mainly smaller than 2.5 $\mu$m and thus the mineralogy is representative of the small particle fraction. In summary, the identified mineralogical composition is well representative for the particle size fraction used for ice nucleation experiments on the Atacama milled and sieved, sieved Etosha, Israel milled, milled Australia, and the airborne Crete, Peloponnese and Tenerife samples.

## 2.5 Immersion freezing experiments and data treatment

Immersion freezing experiments between 235 and 250 K were conducted by extending ZINC (Stetzer et al., 2008) with IMCA (Lüönd et al., 2010). ZINC is a vertically oriented continuous flow diffusion chamber (Rogers, 1988) with two flat parallel walls. The walls are ice-coated before an experiment, and by applying a temperature gradient between the two walls at supercooled temperatures supersaturation with respect to ice is established between the walls. To ensure droplet activation of all sampled particles before freezing, IMCA is installed upstream of ZINC. In IMCA a relative humidity of 120 % with respect to water at a temperature of 303 K is provided by humidified filter paper on the two parallel walls of the chamber. Under these conditions, all particles activate such that each droplet contains a single dust particle. The droplets are then cooled to the experimental temperature before they enter ZINC. For the immersion freezing experiments the relative humidity in ZINC is kept at water saturation. The IODE detector (Nicolet et al., 2010) measures the depolarization signal of a linearly polarized laser beam by the particles. This allows differentiation between spherical droplets, which nominally do not lead to a depolarization signal, and the non-spherical ice crystals which depolarize the laser light. The ratio of the detected ice crystal concentration ($N_i$) to the sum of ice crystals and detected droplet concentration ($N_d$) is called the frozen fraction ($FF$):

$$FF = \frac{N_i}{N_d + N_i}. \tag{1}$$

IODE can distinguish the depolarization signal of droplets and ice crystals between the limits of detection (LOD) of $FF = 0.1$ and $FF = 0.9$. Over the course of about 3 h the temperature is stepwise ramped up. Each data point represents 2000 - 3000 single detected particles.

For independent offline immersion freezing measurements between 250 - 263 K with FRIDGE, dust particles were collected by filtration from the tank over 3.5 h using Teflon membrane filters (Fluoropore PTFE, 47 mm, 0.2 $\mu$m, Merck Millipore Ltd.). The particles were then extracted from the filters into vials with 10 ml of deionized water for 10 minutes in an ultrasonic bath. 150 drops of 0.5 $\mu$l each were randomly placed on a silicon plate on the cold stage of FRIDGE using an Eppendorff-pipette. At ambient pressure conditions the temperature of the cold stage was then lowered by 1 K min$^{-1}$ and the number of drops

freezing as a function of temperature is recorded with a CCD camera. This process is repeated several times with fresh droplets until a minimum of 1000 droplets is exposed. The INP concentration is given by (Vali, 1971; Ardon-Dryer and Levin, 2014)

$$K'(T) = \frac{1}{V_{\text{drop}}} [\ln(N_0) - \ln(N(T))] \frac{V_{\text{water}}}{V_{\text{air}}} \tag{2}$$

where $K'(T)$ is the cumulative INP concentration, $V_{\text{drop}}$ is the volume of a droplet, $N_0$ the number of droplets sampled, $N(T)$ the number of frozen droplets, $V_{\text{water}}$ the volume of water used to wash off the particles from the filter and $V_{\text{air}}$ the volume of air ($N_2$ in the current study) sampled through the filter. The temperature uncertainty is $\pm 0.2$ K and the uncertainty in $FF$ typically $\pm 30$ % at $T \leq 260$ K and decreases with lower temperatures.

Due to the small sample amounts particularly of the airborne dust samples, generating monodisperse particles for the ice nucleation measurements was not possible. Earlier studies have shown that the probability of a particle to act as INP scales with the surface area of the particle immersed in a droplet (Archuleta et al., 2005; Welti et al., 2009; Kanji and Abbatt, 2010). So-called ice-active sites (Vali, 1966) are assumed on the surface of an INP in the deterministic concept (Langham and Mason, 1958). The probability of such a site to be present on a particle increases with the surface area. To compare the $FF$ measured in IMCA from samples with different size distributions, the $FF$ is normalized by the mean aerosol particle surface area. This yields the ice-active surface site density, $n_{\text{s}}$:

$$n_{\text{s}} = -\frac{\ln(1 - FF)}{A_{\text{ve,w}}} \tag{3}$$

In the case of FRIDGE it is calculated as:

$$n_{\text{s}} = -\frac{\ln(1 - \frac{N(T)}{N_0})}{A_{\text{t,drop}}} \tag{4}$$

with $A_{\text{t,drop}}$ being the total aerosol surface area present in each droplet and given as:

$$A_{\text{t,drop}} = \overline{N} V_{\text{air}} \overline{A_{\text{ve,w}}} \frac{V_{\text{water}}}{V_{\text{drop}}} \tag{5}$$

with the mean total aerosol concentration $\overline{N}$ in the reservoir tank during the time of the particle collection as measured by the CPC. The assumption that active sites are uniformly distributed over individual particle surfaces, and therefore that $n_{\text{s}}$ stays constant with particle size, most likely has limitations for complex polymineral samples such as desert dust particles. Therefore, the provided $n_{\text{s}}$ values should not be treated as an exact parameter, valid at any particle size, but rather a normalization method for the bulk natural dust samples $< 2.5 \mu$m which we investigated.

## 3 Results and Discussion

### 3.1 Dust size distribution

Most of the size distributions of the different dust samples in the tank were bimodal. Figure 3 shows exemplary SMPS and APS surface area distribution data of four samples together with the bimodal fit. Since one mode was detected in each instrument's

size range, the shape factor $\chi$ was optimized to give the best overlap of the two size distributions. For any shape factor within realistic limits for atmospheric dusts ($1.1 \leq \chi \leq 1.6$, Alexander 2015) the two modes remained distinguishable. They are likely related to the high inhomogeneity of the samples with respect to hardness and fracture. Two airborne (Crete and Egypt), one surface-collected sieved and one surface-collected milled (both Israel) samples are shown. The Crete sample has a third small mode at $d_{ve} = 50$ nm. Since the smaller aerosol particles contribute only little to the average surface area, the distribution was also bi-modally fitted. The mean particle surface area values are given in Table 1 together with the relative error $\delta(\overline{A_{ve,w}})$ resulting from a change in distribution during the course of the experiment. All samples peak in number concentration between $d_{ve} = 200 - 400$ nm and a mean particle surface area of $\overline{A_{ve,w}} = 2 - 3.7$ $\mu m^2$ corresponding to a diameter of $\overline{d_{ve,w}} = 800\text{-}1100$ nm. Only the Great Basin sample differs strongly because of the presence of predominantly large particles leading to a high mean particle surface area ($\overline{A_{ve,w}} = 16.4$ $\mu m^2$, $\overline{d_{ve,w}} = 2133$ nm). The relative error $\delta(\overline{A_{ve,w}})$ is 64 % in the case of the Great Basin sample because two refills were necessary during the course of the experiment. For all other samples $\delta(\overline{A_{ve,w}})$ is less than 24 %.

## 3.2 Ice nucleation of desert dust

The plots on the left side of Fig. 4 show the $FF$ as a function of temperature between 235 and 253 K, separately for non-Saharan and the Saharan samples. The majority of the non-Saharan $FF$ curves (Fig. 4a and b) behave similarly, with two samples being distinctly different: the Australia sample shows significantly higher $FF$ values at all temperatures whereas the milled Israel sample falls clearly below the other $FF$ curves for all $T > 237$ K. Of the intermediately active samples, the Taklamakan and Great Basin samples are at the upper end whereas the Dubai sample shows the second lowest $FF$ values. The remaining samples are mostly not significantly different, with $FF$ values lying within each others error bars.

The five Saharan samples in Fig. 4c cover a comparable range of $FF$ to the non-Saharan ones at any temperature. The only surface-collected Saharan sample (Morocco) has higher $FF$ values compared to the airborne Saharan samples. All samples were fit with sigmoidal curves. None of the samples shows a stepwise $FF$ owing to the polydisperse size distribution of the particles. Due to the heterogeneous particle composition, a partial step-like activation spectrum could be expected with decreasing temperature if the single mineral components were externally mixed and not present within one particle. Since ice nucleation activity is also dependent on the surface area of each particle (Archuleta et al., 2005; Connolly et al., 2009; Welti et al., 2009), larger particles will activate at higher temperatures than smaller ones, smoothing out the potential step function of different minerals.

$n_s$ was calculated from the $FF$ using $\overline{A_{ve,w}}$ in Eq. 3 to account for differences in the size distributions which may impact the ice nucleation behavior. The results are shown in the plots on the right side of Fig. 4. The error bars in $n_s$ are derived by error propagation from the error in $FF$ and $\delta(\overline{A_{ve,w}})$ and are dominated by the error in $FF$. Data points outside of $0.1 < FF < 0.9$ and in the homogeneous freezing regime are omitted. The $n_s$ of the Australian sample remains the highest of all samples and that of the Israel milled sample one of the lowest. The $n_s$ of the Great Basin sample, which has one of the highest $FF$s, is amongst the lowest, due to its coarse particle sizes. Like their $FF$, the range of $n_s$ of the Saharan samples is comparable to those of the non-Saharan ones (Fig. 4f). Among the Saharan samples, the $n_s$ of the Tenerife sample is similar to that of Crete,

whereas the Egypt sample is higher for $T > 238$ K. The $n_\mathrm{s}$ of the surface-collected and milled Morocco sample is distinctly higher at all temperatures.

To compare all dust samples, the $n_\mathrm{s}$ in m$^{-2}$ was fitted using the exponential function:

$$n_\mathrm{s} = \exp(-a(T - 273.15\mathrm{K}) + b) \tag{6}$$

with the fit parameters $a$ and $b$, which are given in Table 4 for each sample. The resulting fit lines from all samples are shown in Fig. 5a. Overall the Australian sample is by far the most ice nucleation-active sample. The Israel milled, the Great Basin, and the Peloponnese sample show a low ice nucleation activity. For comparison, the $n_\mathrm{s}$ fits for K-feldspar from Atkinson et al. (2013), for kaolinite KGb-1b from Murray et al. (2011), for Illite NX from Broadley et al. (2012), data for quartz from Atkinson et al. (2013) and Zolles et al. (2015) and the $n_\mathrm{s}$ parameterization curve from Niemand et al. (2012) are shown. The

K-feldspar, kaolinite, illite and quartz curves and data points were provided as $n_\mathrm{s,BET}$, i.e. the surface area of the particles was measured with the Brunauer, Emmett and Teller (BET) nitrogen adsorption method (Brunauer et al., 1938). This method yields typically a higher surface area than that based on volume equivalent diameter. The literature $n_\mathrm{s,BET}$ was converted to $n_\mathrm{s}$ using a conversion factor of 3.5 in case of K-feldspar as given in the supplementary material of Atkinson et al. (2013). For illite, we followed Hiranuma et al. (2015), using a specific surface area ($SSA$) of 104.2 m$^2$ g$^{-1}$ (Broadley et al., 2012) and a ratio of

total surface area to total mass of 6.54 m$^2$ g$^{-1}$ (Hiranuma et al., 2015). Similarly, for kaolinite we used $SSA = 11.8$ m$^2$ g$^{-1}$, a density of 2.63 g cm$^{-3}$ and a mean mass-weighted diameter of 674 nm (Hudson et al., 2008), yielding a correction factor of 3.49. For the same $n_\mathrm{s,BET}$ values, the three quartz samples from Zolles et al. (2015) were active over a range of 10 K. No $SSA$ values were provided therefore we used also a conversion factor of 3.5 given that feldspar is somewhat similar to quartz. This comes with a very high uncertainty, as the size distribution and particle shape of quartz are likely to differ from the K-feldspar

of Atkinson et al. (2013). The K-feldspar, kaolinite, illite and quartz $n_\mathrm{s}$ areas cover the range from the $n_\mathrm{s,BET}$ as provided in the literature and the calculated $n_\mathrm{s}$ to show the uncertainty inherent to the conversion. It can be seen that all desert dust samples fall between the K-feldspar and the clay mineral and quartz fits at all temperatures.

Similarly to our study the parameterization from Niemand et al. (2012) was based on the $n_\mathrm{s}(T)$ of three polydisperse surface-collected dust samples from China, Egypt and the Canary Islands and one sample collected after deposition in Israel. For

$T < 250$ K the parameterization falls in the lower end of the range of $n_\mathrm{s}$ observed for our broader collection of global surface-collected and airborne dust samples (a factor of 3 to 4 below the average $n_\mathrm{s}(T)$ of all measured curves, not shown). Given that the measurements were conducted with different instruments, which can lead to a systematic offset of up to three orders of magnitude in terms of $n_\mathrm{s}$ (Hiranuma et al., 2015), and the polydisperse size distribution of the dust samples, the agreement is considered reasonable. The maximum difference in temperature between the parameterization from Niemand et al. (2012) and

the average of all $n_\mathrm{s}$ curves from this study (not shown) is less than 3 K, while the spread in $n_\mathrm{s}$ curves across all samples in our study is up to 10 K. The parameterization has a slope close to that of all airborne samples, whereas most of the surface-collected samples show a more moderate slope, i.e. a lower temperature dependence. This can be seen as an indication of active sites, which activate at warmer temperatures, being more frequent in the surface-collected samples compared to the airborne samples.

The high temperature measurements from FRIDGE for four samples are shown in Fig. 5b together with those from IMCA and parameterizations. Again, the desert dusts fall between extrapolations of the clay mineral and K-feldspar fits. The parameterization from Niemand et al. (2012) predicts one to two orders of magnitude higher $n_s$ than measured by FRIDGE. Only for the Atacama milled sample at $T = 251$ - 256 K the parameterization shows about 30 % higher values than the measurements. While at $T < 250$ K the $n_s$ values of the four samples mostly overlap within error bars, at $T > 250$ K the Atacama milled sample has about one order of magnitude higher $n_s$ than the other three samples. This shows that the fits of $n_s(T)$ are not constant over the whole temperature spectrum and are only valid for the given range. It indicates that the Atacama milled sample contains active sites at these temperatures which are missing in the other samples.

Figure 6a shows the median and minimum to maximum $n_s(T)$ range of the airborne and surface-collected samples. This illustrates that the $n_s$ range of the airborne samples falls in the lower half of the $n_s$ range of the surface-collected samples or even below. It shows that for immersion mode ice nucleation surface-collected dust samples are not representative for airborne dust samples, which all stem from North Africa, the world's largest source of atmospheric dust. This might be caused by a non-representative surface-dust collection, e.g. soil rather than dust is collected which has a different size distribution and composition, or dust from a location where threshold wind velocities for dust lifting are not reached. Another cause could be that atmospheric processes taking place during or after particle lofting may alter the particle surface and decrease the ice nucleation ability which has been suggested to occur in the field (Cziczo et al., 2013) and laboratory (Sullivan et al., 2010; Augustin-Bauditz et al., 2014). The potential effects of mineralogy on the ice nucleation activity at different temperatures is investigated in the following section.

## 3.3 Role of mineralogy

Various earlier studies have shown that the ice nucleation activity expressed by $n_s$ varies by several orders of magnitude between different types of minerals. By analyzing the bulk mineralogy we investigate if the dust's mineralogical composition explains the observed ice nucleation activity. Tables 2 and 3 show the results of the mineralogical analysis of the dust samples. The distinct composition of the Australia sample is striking, consisting almost entirely of quartz (91.3 wt%) and K-feldspars (4.2 wt% orthoclase, 3.9 wt% microcline), which are highly ice-active minerals in the immersion mode (Atkinson et al., 2013; Zolles et al., 2015). The Morocco sample also has a high quartz content (63.8 wt%), followed by the Taklamakan sample (33.1 wt%), both being two of the most ice-active samples of this study. The remaining samples have a quartz content of 23 wt% or less. Another obvious difference is the high feldspar content of both Atacama samples (milled: 22.3 wt% orthoclase, 43.2 wt% Na-plagioclase, sieved: 11.8 wt% orthoclase, 39.3 wt% Na-plagioclase). The milled Atacama sample shows the highest $n_s$ of the four investigated samples in FRIDGE at $T > 250$ K (Fig. 5b), close to the K-feldspar parameterisation by Atkinson et al. (2013) at $260 < T < 262$ K and also higher activities than the sieved sample at $T > 240$ K (Fig. 6b). The Israel samples have a distinctly high calcite content (milled: 81 wt%, sieved: 67.2 wt%), followed by Dubai (37.2 wt%) and Peloponnese (33 wt%). Calcite has been found to be a weakly ice-active mineral in the immersion mode (Atkinson et al., 2013; Zolles et al., 2015) and also in the condensation mode (Roberts and Hallett, 1969; Zimmermann et al., 2008). The Etosha

sample consists of about one quarter each of calcite (29 wt%), dolomite (27 wt%) and ankerite (23 wt%) with 10 wt% muscovite and no significant fraction of clay minerals, feldspars or quartz (1 wt% smectite and 1 wt% quartz were identified). This is surprising as the Etosha sample is one the most ice nucleation-active samples at $T < 242$ K. The ice nucleation ability of the mica muscovite is debated as some studies have found hardly any ice nucleation activity at heterogeneous freezing tem-

peratures (Atkinson et al., 2013; Campbell et al., 2015; Kaufmann et al., 2016) while others found significant ice nucleation ability at $T < 243$ K (Steinke, 2013; Abdelmonem et al., 2015). Kaufmann et al. (2016) found little ice nucleation activity of a reference dolomite sample. Thus, the high ice nucleation activity at $T < 242$ K of the Etosha sample is not explainable by the known ice nucleation ability of its mineral components. To our knowledge, no study so far has investigated the ice nucleation behavior of pure ankerite.

The remaining samples are more complex mixtures of quartz, feldspars, clay minerals, micas and other minerals. We find less than 4 wt% microcline in the sieved and milled Israel and the airborne Tenerife samples. In the samples from Australia and Morocco both K-feldspars orthoclase and microcline seem to be present. The surface-collected Great Basin sample contained 30 wt% microcline in the bulk sample but likely much less in the size fraction $< 2.5$ $\mu$m. The Saharan samples show a great variety with the Tenerife sample having the highest content of clay minerals (illite + kaolinite + smectite + palygorskite:

57.7 wt%) of all dusts, similar to the findings of Alastuey et al. (2005). The other airborne Saharan samples (Egypt, Peloponnese and Crete) consist to about 50 wt% of quartz and calcite, likely due to different source regions within the Sahara. The mineral fraction of the main minerals in soil samples is not homogeneous throughout the Sahara (Nickovic et al., 2012). Based on air mass back trajectory calculations the Tenerife sample originated in Northern Mauritania or Morocco, whereas the Egypt sample stemmed from local sources in Egypt and the Peloponnese and Crete samples from Northern Saharan sources

in Algeria, Tunisia and Libya. Schütz and Sebert (1987) found that the mineral composition of the size fraction $< 5$ $\mu$m of surface-collected samples in the Sahara was very similar throughout the Sahara and concluded that this size fraction is already well mixed within the desert. The main differences were a higher calcite and palygorskite content in the Northern Sahara which is consistent with our findings for the Egypt, Peloponnese and Crete samples in the case of calcite. We find comparable amounts of palygorskite in the Crete, Peloponnese and Tenerife samples (4.5 - 5.3 wt%) but no palygorskite in the Egypt

and Morocco samples. The differences with regard to mineralogical composition and ice nucleation ability found between the Morocco (higher quartz content) and Tenerife (higher clay mineral content) samples shows that, even if the source region of an airborne sample can be roughly localized, the mineralogy of surface-based and airborne samples can differ: surface-collected samples are not necessarily representative of the dust aerosol. This is supported by the fact that the Morocco sample consisted mostly of dust grains larger than 32 $\mu$m, which sediment quickly before being transported long distances in the atmosphere.

In the following we investigate if the different $n_{\mathrm{s}}$ values of the dust samples can be attributed to their mineralogy. For this, we compare the fraction of single minerals to the $n_{\mathrm{s}}$ of our dust samples at five different temperatures, using the most prominent minerals which have been found to have $n_{\mathrm{s}}$ values in a range which is measurable with IMCA-ZINC (Atkinson et al., 2013; Hiranuma et al., 2015; Zolles et al., 2015; Harrison et al., 2016; Peckhaus et al., 2016). Those are quartz, illite, kaolinite, and calcite. Additionally, we compare the samples' $n_{\mathrm{s}}$ to the following sums of minerals: K-feldspars (microcline plus orthoclase);

all feldspars (K-feldspars plus Na-plagioclase feldspars); sum of all feldspars plus quartz; sum of all feldspars plus quartz plus

illite and/or kaolinite. We do not differentiate between the different K-feldspar polymorphs because even the same type of feldspar can vary in $n_s$ as shown by Harrison et al. (2016) due to the complex structure of feldspars. For each temperature, all samples which showed a $FF$ between 0.1 and 0.9 were used for the correlations. The Etosha sample was excluded from the correlations with feldspars, quartz and clays, as it does not contain any significant amount of these minerals. However, it was included for the comparison with calcite.

Table 5 shows the Pearson correlation coefficients ($R$) between the mineral fractions and the $n_s$ at five temperatures. Only a few correlations are statistically significant, owing to the low number of samples. Nevertheless, the overview of the correlation coefficients gives an idea of the effect of certain minerals on $n_s$. The related scatter plots are given in Figure 1 of the supplementary material.

At 253 K only three samples (Atacama milled, Egypt and Taklamakan) are available for comparison. For these, the K-feldspar content leads to a very high correlation ($R = 0.97$) and adding the Na-plagioclase to the feldspar sum reduces the $R$ value to 0.87. At lower temperatures, no correlation is found between the K-feldspar content and $n_s$. At 245 K, the $n_s$ correlates best ($R = 0.89$) with the quartz alone and adding feldspar, illite or kaolinite leads to a lower $R$ value (0.65 - 0.80). This is in agreement with earlier studies showing the comparable high immersion mode ice nucleation activity of some quartz samples at this temperature (Atkinson et al., 2013; Zolles et al., 2015). Interestingly, this behavior stays the same at 243 K ($R = 0.91$). At 240 K and 238 K the quartz correlation is not as good and adding feldspars to the quartz improves the correlation (from $R = 0.52$ and 0.45 to 0.65 and 0.54, respectively). Note that the Australia sample, which has the highest quartz content, is excluded at 240 K and 238 K from the correlation analysis because $FF$ was $> 0.9$. Illite alone shows a weak negative correlation with $n_s$ at any of the investigated temperatures and leads to reduced or constant $R$ values when added to the quartz plus feldspar sum. Also calcite and kaolinite are negatively correlated with $n_s$ at all presented temperatures. This means that a higher ice nucleation activity in one sample can be attributed at 253 K to the K-feldspar content present in this sample whereas at temperatures between 238 - 245 K it is attributed to the sum of feldspar and quartz content present. A high clay mineral content on the other hand is associated with lower ice nucleation activity. To exclude a bias from varying numbers of samples at different temperatures, we have repeated the correlations at 245, 243, 240 and 238 K for the Atacama milled, Egypt and Taklamakan samples only. The corresponding scatter plots are provided as Figure 2 of the supplementary material and confirm the observations that quartz plus feldspar yield the best correlations at $T < 245$ K.

These results need to be treated carefully, because the mineralogy is derived from the full size range up to 32 $\mu$m and may be different to the studied size fraction ($< 2.5$ $\mu$m). Therefore, we do the same analysis exclusively for the samples for which the size fraction larger than 2.5 $\mu$m was small and the mineralogical composition determined by XRD can be assumed to be representative for particles $< 2.5$ $\mu$m. These samples are: the Atacama milled and sieved samples, the sieved Etosha, milled Israel and Australia, and the airborne Crete, Peloponnese, and Tenerife samples. From these samples only the ice nucleation ability of the Atacama milled and the Etosha sample was measured with FRIDGE at 253 K and only two of the samples show measurable $n_s$ at 245 K in the IMCA data, thus those temperatures are excluded. The results for 243 K, 240 K, and 238 K are given in Table 6. The correlation with illite and kaolinite is still negative at any temperature despite the fact that some of the samples in this subset contain a comparably large amount of these clay minerals (Tenerife: 21.8 wt%, Peloponnese: 20.2 wt%).

At 243 K quartz shows the highest $R$ value (0.91) for the selected samples. At 240 K and 238 K the sum of all feldspars ($R = 0.95$ and $R = 0.88$) and the sum of all feldspars plus quartz ($R = 0.97$ and $R = 0.90$) show the highest correlation with $n_\mathrm{s}$. This is different to when all samples were included, where quartz exclusively led to the highest $R$ values. The difference stems mainly from the exclusion of the Morocco, Taklamakan, Egypt and Great Basin samples, which all had a high quartz content.

It should be noted that XRD is a bulk analysis of the mineralogical composition whereas ice nucleation is a process sensitive to the particle's surface, including cracks, crevices or pores (Marcolli, 2014; Welti et al., 2014). Despite this difference in sensitivity, a high correlation between bulk mineralogy and ice nucleation activity is found in this study. As XRD does not allow any inference of the mixing state of different minerals it is not known, particularly for the small particles, if each particle contains some amount of quartz and/or feldspar or if pure calcite or clay mineral particles exist. However, our results indicate

that feldspar or quartz present in the bulk dust will dominate its freezing behavior down to 238 K. While the correlations do not exclude an influence of non-mineral material, the results suggest that potential coatings or mixing of the particles only play a secondary role for the immersion freezing ability at the studied temperatures of the mineral dusts. The majority of the results can be explained by the mineralogy, as also observed by Kaufmann et al. (2016). This may be due to a significant dilution of coating material in the droplets forming on each dust particle and may have a more prominent role in deposition nucleation.

Measurements in the condensation mode, which is the subject of part 2 of this study, suggest that the ice nucleation activity of this sample is in large part related to organic or biological material mixed with the dust.

### 3.4  Effect of milling

Two of the dust samples have undergone two different treatments (sieving and milling) to compare the effect of milling on $n_\mathrm{s}$ of

a polymineral sample. The Israel sample was first sieved and then part of the $d \leq 32\ \mu$m fraction was milled. For the Atacama sample, the original sample containing particles of all sizes was split, one part was sieved and one part milled. The resulting $n_\mathrm{s}$ curves are shown in Fig. 6b. Interestingly, the two sieved samples are very similar in $n_\mathrm{s}$. The milling of the Atacama sample led to a slightly higher ice nucleation efficiency at $T > 240$ K compared to the sieved sample. Contrastingly, for the Israel sample milling led to a decrease in $n_\mathrm{s}$ at all studied temperatures above 237 K. The latter could be related to the high calcite content

of the Israel dust. Calcite is a rather soft mineral with a Mohs hardness of 3 and a perfect cleavage, and during the milling process it could be ground to a smaller grain size faster than compared to harder minerals such as quartz (Mohs hardness of 7) or feldspar (6). Thus the size fraction $d_\mathrm{ve} \leq 2.5\ \mu$m (the $D_{50}$ cut-off of the particle generation system used) could be enriched in calcite. Calcite has been found to be ice-active only very close to the homogeneous freezing regime (Atkinson et al., 2013; Zolles et al., 2015) and negatively correlated with $n_\mathrm{s}$ at all presented temperatures in this study (Tables 5 and 6). Similarly, the

slightly higher $n_\mathrm{s}$ at $T > 240$ K of the milled Atacama sample is likely due to more feldspar being present in the $d_\mathrm{ve} > 32\ \mu$m fraction compared to the sieved sample, which then got milled into sizes $d_\mathrm{ve} < 2.5\ \mu$m. It can be seen from Tables 2 and 3 that the K-feldspar (orthoclase) content is higher in the Atacama milled sample (22 wt%) than in the sieved one (12 wt%). The milled sample mostly consisted of particles smaller than 2.5 $\mu$m, whereas the sieved one had a large fraction of particles larger than 2.5 $\mu$m. Thus, there was likely more orthoclase content in the milled Atacama sample particles smaller than 2.5 $\mu$m

compared to those in the sieved sample, leading to higher $n_\mathrm{s}$ at warmer temperatures. Effects such as an increase in surface irregularities, defect density and functional groups due to the milling as reported by other authors (Hiranuma et al., 2014; Zolles et al., 2015), are not excluded but were not investigated in this study. An increase in defect density and surface irregularities has been shown to increase the ice nucleation activity of monomineral or single compound samples. As milling reduced the ice

nucleation activity of the Israel sample, we conclude for this specific sample that any morphology effect is small in comparison to the change in mineralogical composition of the analyzed size range caused by the milling. This emphasizes the importance of mineralogy for the surface sensitive ice nucleation process.

## 4   Conclusions

The ice nucleation ability in the immersion mode of 15 natural desert dust samples was quantified by the frozen fraction and ice-active surface site density, and compared with the bulk dust mineralogy. A diverse mineralogical composition was found for the different desert dust samples which can be related to variable ice nucleation abilities. The comparison showed that at temperatures above 250 K the highest $n_\mathrm{s}$ is related to the highest K-feldspar content in the sample confirming earlier findings on the superior ice nucleation ability of K-feldspars compared to other minerals. Microcline was found in one airborne sample

(4 wt%) and in surface-collected samples from four different locations. We could not confirm a superior role of microcline over orthoclase, in-part because a differentiation of the two minerals was often difficult and partly because their content was too low in the size range investigated for ice nucleation to cause an effect detectable with the IMCA-ZINC experiment. A conclusion on the atmospheric relevancy of microcline is therefore not possible because even in low amounts - of a few percent - it could nucleate ice and glaciate clouds at temperatures warmer than 253 K.

At temperatures below 250 K, the ice nucleation ability was mainly attributed to the quartz content of the samples, as well as to the sum of all feldspars (K-feldspars and Na-plagioclase feldspars). Keeping in mind that quartz is ubiquitous in atmospheric desert dust, this suggests that quartz plays a more prominent role for atmospheric ice nucleation than previously thought. The clay mineral (illite and kaolinite) and calcite content of the dust samples negatively correlated with $n_\mathrm{s}$ at all studied temperatures, suggesting a minor importance of these minerals for the ice nucleation activity of natural dust samples in the immersion

mode, especially if quartz or feldspar are present. Atkinson et al. (2013) suggested that the global mineral dust INP concentration down to a temperature of about 240 K is dominated by feldspar. At temperatures between the homogeneous freezing limit and 240 K, where quartz is an active INP, it dominates the total INP concentration as it is much more abundant than feldspar. Our experiments on natural dust confirm this suggestion.

   The variation in mineralogy with particle size leads to variations in $n_\mathrm{s}$. The most ice nucleation-active mineral feldspar is less

common in the smallest dust size fractions. Quartz is found in all size ranges but more common in the larger dust size fractions. The less ice nucleation-active clay minerals and calcite dominate the small size fraction. As the size distribution of dust changes during atmospheric transport, a resulting change in mineralogy will have implications for the atmospheric relevance of certain mineral components. Three of the four airborne samples (Crete, Peloponnese and Tenerife) had the highest clay mineral

content and were amongst the least ice nucleation-active samples. For all desert dust samples we found a high correlation of the ice nucleation activity of particles smaller than 2.5 $\mu$m with the quartz content of the dust samples. This shows that despite the dominance of the clay minerals in the small size fraction, quartz is an important atmospheric INP component and also found in the particle size fraction with the longest atmospheric residence time.

Milling of dust samples in the laboratory or by natural mechanical weathering processes can lead to more surface inhomogeneities and also to an enrichment of more ice nucleation-active minerals, such as quartz, in sizes relevant for atmospheric ice nucleation. This potentially explains the higher ice nucleation activity of Arizona test dust compared to desert dust samples found in earlier studies (Zolles et al., 2015; Kaufmann et al., 2016). The observed differences between the surface-collected and the airborne Saharan dust samples suggest that surface-collected dust may not be representative of atmospheric dust trans-

ported over long distances. Thus, more ice nucleation studies on airborne, transported dust also from deserts other than the Sahara are crucial to quantify the ability of atmospheric dust to nucleate ice. Furthermore, if airborne dust is generally found to be less ice nucleation-active than surface-collected dust and does not show ice nucleation activity at temperatures above 263 K, it cannot explain first ice formation in clouds with top temperatures warmer than 263 K. For that other INP such as biological particles or different mechanisms such as seeder - feeder processes would need to be investigated. In our study the

ice nucleation activity of only one sample (Etosha) was not explainable by the known ice nucleation activity of its mineral components. The role of adsorbed organic material for the ice nucleation activity of the dust samples will be investigated in part 2 of this paper series.

The applicability of the parameterization by Niemand et al. (2012) to describe an average ice nucleation behavior of desert dust was confirmed by a global set of dust samples. However, the variation between the different samples in temperature was

up to 10 K. To more adequately describe immersion freezing by desert dust in the atmosphere, mineralogy sensitive emission and transport schemes would be desirable. We suggest, K-feldspar for temperatures above 250 K and additionally at lower temperatures Na-plagioclase feldspars and quartz emissions and transport should be quantified. Since this is complex and computationally expensive to implement, more studies quantifying the ice nucleation ability of dust as it is found in the atmosphere may circumvent this complexity.

## 5  Data availability

The ice nuclation data shown in Fig. 4 and the data used for the correlation coefficient analysis are uploaded here: https: //polybox.ethz.ch/index.php/s/CU6iHJR0ITGFpcb. Other data can be obtained from the corresponding authors.

*Author contributions.*  YB collected the Tenerife and Israel samples, initiated, planned and led the measurements, performed and analyzed

the XRD measurements, analyzed the INP and aerosol data and wrote the manuscript, AW, JA and FR conducted the IMCA measurements, AW analyzed the IMCA data and collected the Australia, Mojave and Great Basin sample, AD performed the FRIDGE sample collection and

analysis, HB supervised the FRIDGE data analysis and contributed to the manuscript, MP performed and analyzed the XRD measurements, BS, ZAK and UL supervised the project and contributed to the manuscript.

*Acknowledgements.*  The various dust samples in this paper have been collected by a number of people who the authors are very thankful to: Maria Kanakidou and her team (Peloponnese, Crete), Felix Lüönd (Atacama), Paolo D'Odorico and Christopher Hoyle (Etosha), Lukas Kaufmann, Konrad Kandler and Lother Schütz (Taklamakan), Monika Kohn (Dubai), Joel Corbin (Morocco), Sergio Rodríguez (Tenerife) and unknown Egyptian researchers (Egypt). Furthermore, we thank the rest of campaign members: Annika Kube, Larissa Lacher and Fabian Mahrt, as well as Hannes Wydler for technical support. Y. Boose is funded by the Swiss National Science Foundation (grant 200020 150169/1). The research leading to these results has received funding from the European Union's Seventh Framework Programme (FP7/2007-797 2013) under grant agreement n° 603445 (BACCHUS). AD and HB gratefully acknowledge support by Deutsche Forschungsgemeinschaft (DFG) under the Research Unit FOR 1525 (INUIT).

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

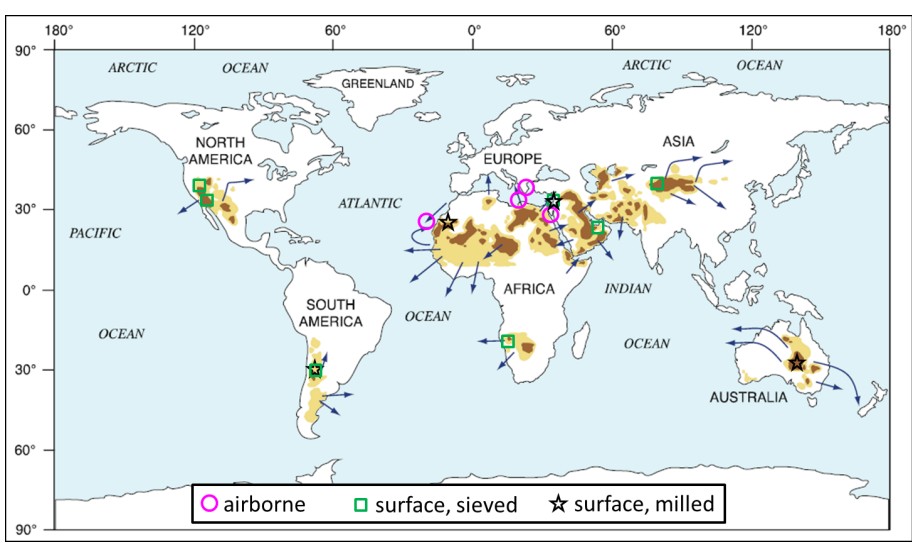

**Figure 1.** Collection sites of the dust samples. Green squares/black stars indicate sieved/milled samples which were collected directly from the surface, pink circles indicate samples that were collected either directly from the air or by deposition after transport from the Sahara. See text for details on the collection methods and treatment after collection. The map was adapted from Knippertz and Stuut (2014) and is based on data from Total Ozone Mapping Spectrometer (TOMS) satellite data of the absorbing aerosol index ($AAI$). Dark brown color indicates 21 - 31 days of $AAI > 0.7$, corresponding with significant amounts of dust or smoke. Yellow indicates 7 - 21 days of $AAI > 0.7$. Arrows show typical dust transport pathways in the atmosphere.

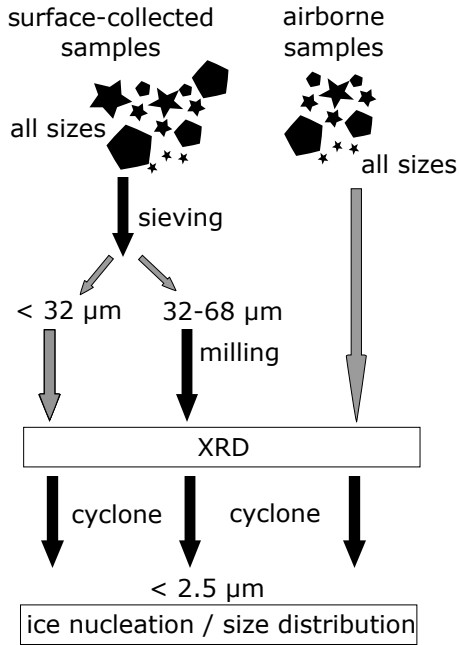

**Figure 2.** Schematic of size fractions used for XRD and ice nucleation.

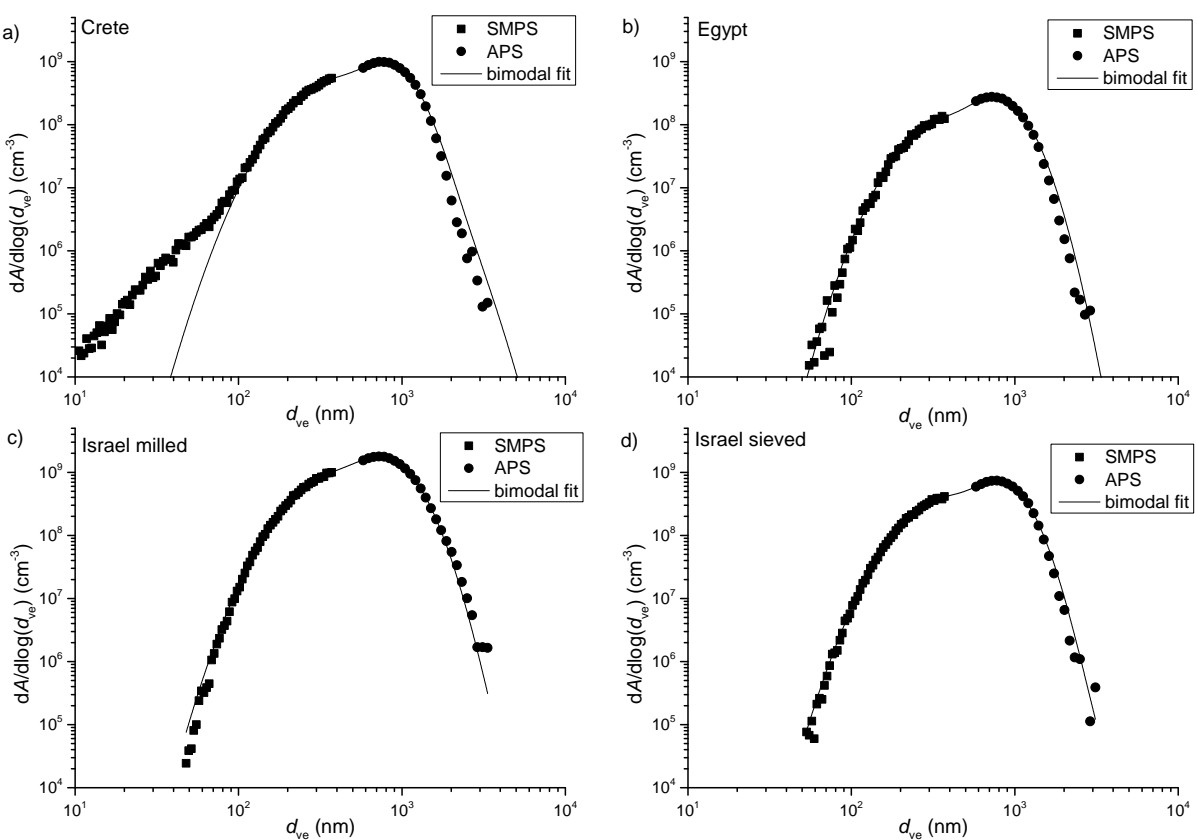

**Figure 3.** Sample surface area distribution of four of the dust samples with bimodal fits.

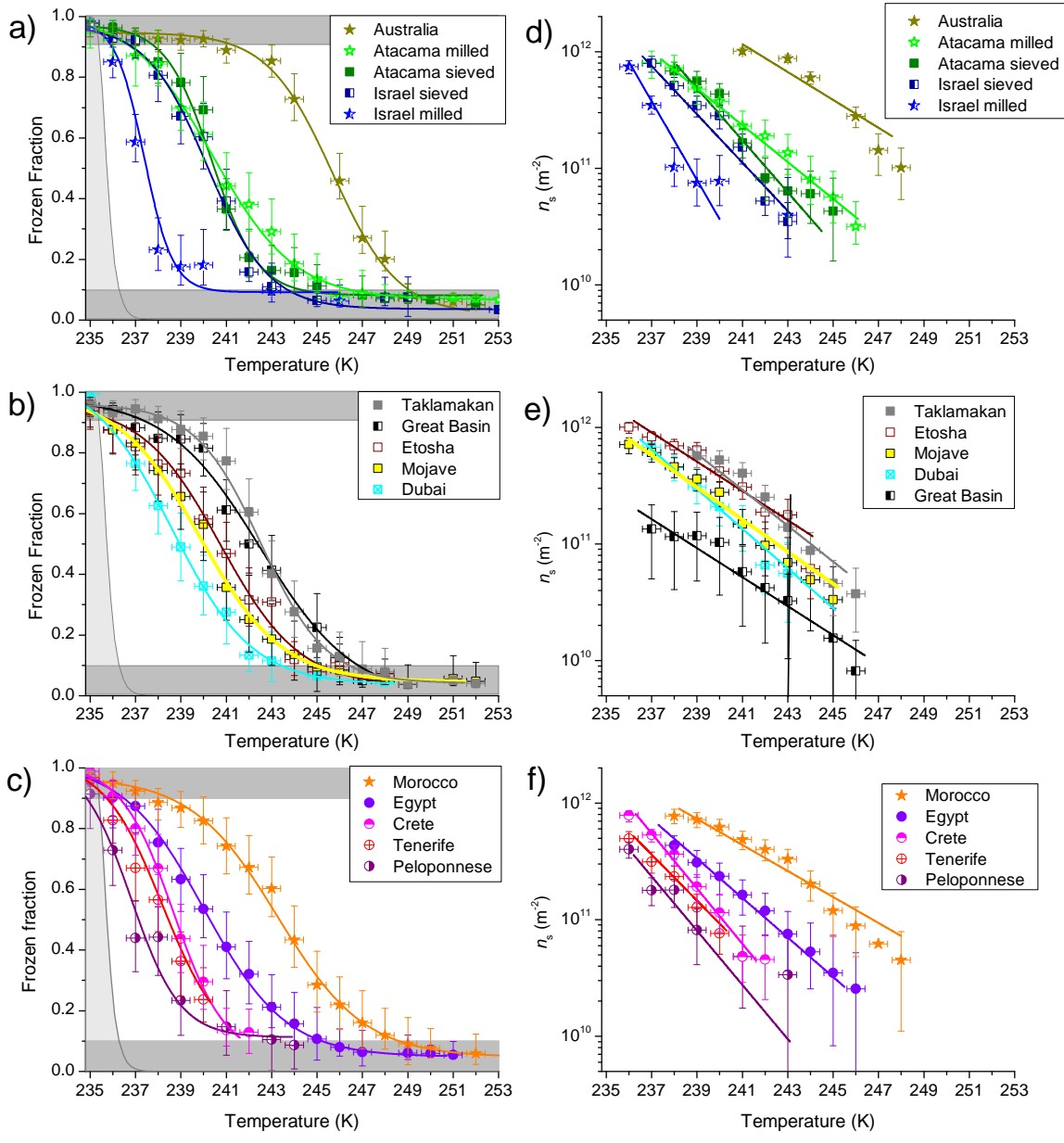

**Figure 4.** Frozen fraction of non-Saharan samples (panels a and b) and c) samples originating in the Sahara. Data were binned into 1 K-intervals. Lines are best sigmoidal fits. Squares: surface-collected and sieved, stars: surface-collected and milled and circles: airborne samples. The light gray area is the homogeneous freezing regime derived from classical nucleation theory (Hoyle et al., 2011; Ickes et al., 2015) and the two dark gray rectangles show the upper and lower detection limits of IODE. Ice-active surface site density of non-Saharan (panels d and e) and f) Saharan samples. Lines are best exponential fits to Eq.6. The fit parameters are given in Table 4.

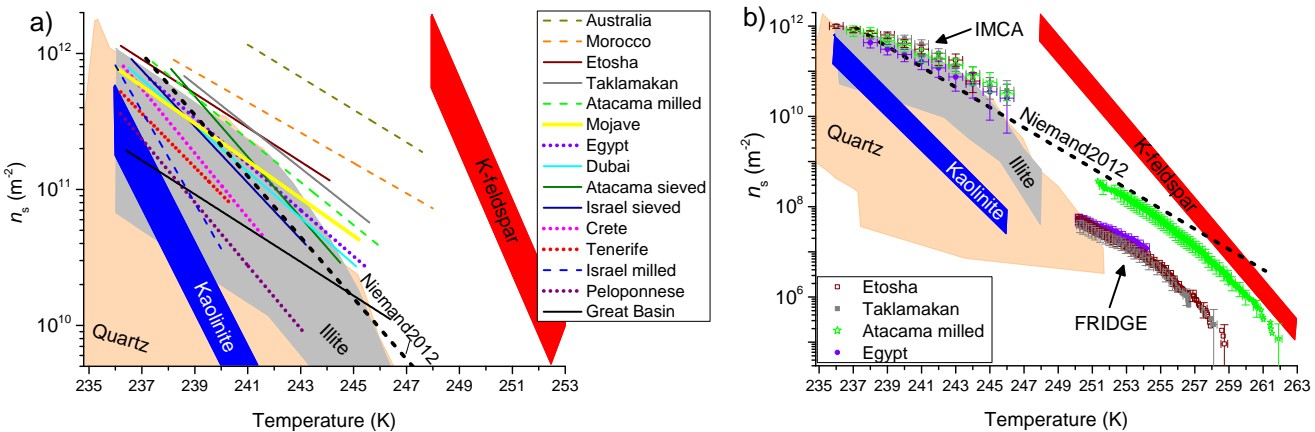

**Figure 5.** a) Ice-active surface site density fits for all dust samples. The fit parameters are given in Table 4. Solid lines indicate surface-collected and sieved samples, dashed lines surface-collected and milled samples and dotted lines are airborne samples. b) Full temperature range measurements taken by IMCA and FRIDGE. Colors and markers are the same as in Fig. 4. IMCA data are binned into 1 K-intervals. Error bars are drawn for every 10[th] data point. A parameterizations for desert dust from Niemand et al. (2012) is shown as thick dashed line. Parameterizations for kaolinite (Murray et al., 2011), illite (Broadley et al., 2012) and K-feldspar (Atkinson et al., 2013) as well as the range of data for quartz (Atkinson et al., 2013; Zolles et al., 2015) are given as areas representing the range between $n_{s,BET}$, as provided in the literature, and the corresponding $n_{s,geo}$. See text for details on the calculations of $n_{s,geo}$.

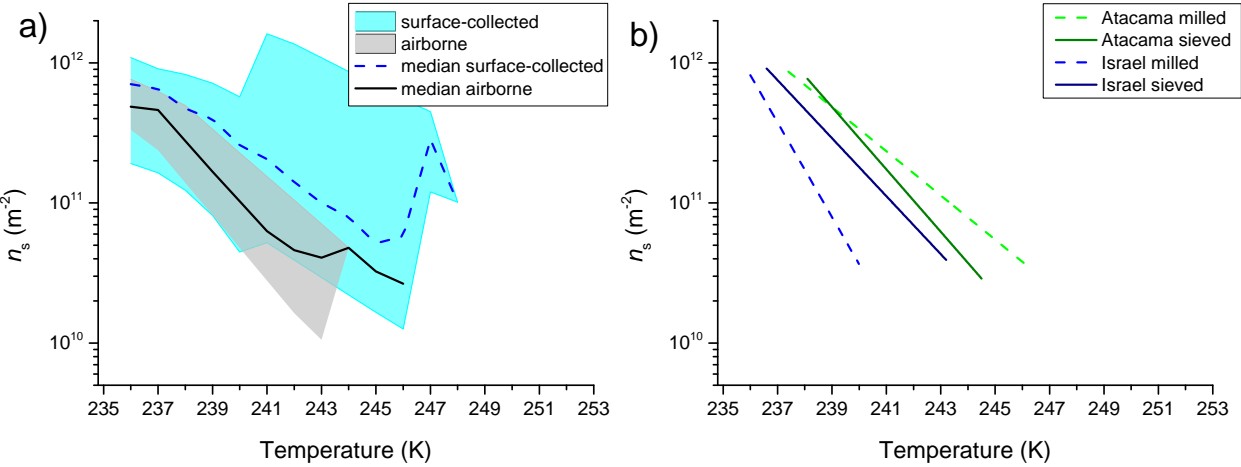

**Figure 6.** a) Median and minimum to maximum $n_s$-range of airborne and surface-collected samples. b) Comparison of ice-active surface site density of milled and sieved samples.

**Table 1.** Overview of the dust size distribution parameters: the mean particle surface area per particle $\overline{A_{\text{ve,w}}}$ with relative error $\delta(\overline{A_{\text{ve,w}}})$ and the corresponding diameter of a particle with this surface area: $\overline{d_{\text{ve,w}}}$ with the relative error $\delta(\overline{d_{\text{ve,w}}})$.

| Sample number | collection site | type | $\overline{A_{\text{ve,w}}}$ ($\mu$m$^2$) | $\delta(\overline{A_{\text{ve,w}}})$ | $\overline{d_{\text{ve,w}}}$ (nm) | $\delta(\overline{d_{\text{ve,w}}})$ |
|---|---|---|---|---|---|---|
| 1 | Atacama | sieved | 2.79 | 0.17 | 940 | 0.08 |
| 2 | Atacama | milled | 2.56 | 0.24 | 897 | 0.11 |
| 3 | Australia | milled | 2.14 | 0.09 | 824 | 0.05 |
| 4 | Crete | airborne | 3.04 | 0.08 | 983 | 0.04 |
| 5 | Dubai | sieved | 2.18 | 0.18 | 830 | 0.09 |
| 6 | Egypt | airborne | 3.26 | 0.12 | 1017 | 0.06 |
| 7 | Etosha | sieved | 2.08 | 0.07 | 813 | 0.03 |
| 8 | Great Basin | sieved | 16.4 | 0.64 | 2133 | 0.39 |
| 9 | Israel | sieved | 3.32 | 0.14 | 1024 | 0.07 |
| 10 | Israel | milled | 2.57 | 0.12 | 904 | 0.06 |
| 11 | Mojave | sieved | 2.99 | 0.14 | 973 | 0.07 |
| 12 | Morocco | milled | 2.81 | 0.13 | 960 | 0.07 |
| 13 | Peloponnese | airborne | 3.27 | 0.06 | 1020 | 0.03 |
| 14 | Taklamakan | sieved | 3.69 | 0.19 | 1079 | 0.09 |
| 15 | Tenerife | airborne | 3.55 | 0.12 | 1061 | 0.06 |

**Table 2.** Mineralogical composition in wt% and uncertainty from the Rietveld refinement (see text for details). Where microcline and orthoclase are present in the same sample, their individual fraction could not be distinguished reliably. Nevertheless, the best Rietveld fit results are given for orthoclase and microcline individually in these cases. The Etosha sample was taken from Kaufmann et al. (2016) who do not provide a Rietveld fit uncertainty from AutoQuan but estimate a 15 % accuracy.

| Mineral Type | Atacama sieved | Atacama milled | Australia milled | Crete airborne | Dubai sieved | Egypt airborne | Etosha sieved | Great Basin sieved |
|---|---|---|---|---|---|---|---|---|
| Ankerite | | | | | | | 23 | |
| Biotite | | 2.8±0.6 | | | 1.0±0.5 | | | |
| Calcite | | | | 25.0±0.6 | 37.2±0.6 | 29.2±1.2 | 29 | 12.9±0.4 |
| Chlorite | | | | 3.8±0.6 | 14.0±1.2 | 8.2 ±1.3 | | 2.1 ±0.3 |
| Cristobalite | 12.7±1.9 | 14.0±1.7 | | | | | | |
| Dolomite | | | | 3.1±0.7 | 6.8 ±0.5 | 8.1 ±1.3 | 27 | 1.9±0.3 |
| Gypsum | | | | 3.7±0.7 | | 5.5±0.9 | | 2.8±0.3 |
| Halite | | | | 1.0±0.2 | | 4.4±0.5 | | 2.1±0.1 |
| Hematite | 4.0±0.5 | 3.2±0.4 | 0.6±0.1 | 0.9±0.2 | 1.4±0.2 | | | 0.9±0.2 |
| Hornblende | 1.3±0.7 | 1.8±0.6 | | 1.5±0.4 | 1.8±0.5 | | | 1.0±0.5 |
| Illite | 10.0±1.0 | | | | | | | |
| Kaolinite | | | | 12.4±1.0 | | 10.5±1.5 | | 17.7±0.9 |
| Microcline | | | 3.9±0.5 | | | | | 30.1±0.8 |
| Muscovite | 4.2±1.0 | 2.4±0.6 | | 9.0±0.6 | 4.1±0.5 | 7.6±1.3 | 10 | 4.0±0.5 |
| Orthoclase | 11.8±0.9 | 22.3±0.9 | 4.2±0.5 | 5.1±0.6 | 2.4±0.4 | 3.5±1.1 | | |
| Palygorskite | | | | 4.5±0.5 | 3.4±0.5 | | | |
| Na-plagioclase | 39.3±1.6 | 43.2±1.4 | | 7.2±0.4 | 9.5±0.4 | | | 3.7±0.3 |
| Smectite | | | | | | | 1 | |
| Quartz | 16.7±0.6 | 10.4±0.4 | 91.3±0.5 | 23.0±0.5 | 13.3±0.3 | 23.0±1.1 | 1 | 20.1±0.5 |
| others | | | | | 5.1±0.4 | | 9 | 0.7±0.3 |

Table 3. Mineralogical composition in wt% and uncertainty from the Rietveld refinement using AutoQuan continued. Where microcline and orthoclase are present in the same sample, their individual fraction could not be distinguished.

| Mineral Type | Israel sieved | Israel milled | Mojave sieved | Morocco milled | Peloponnese airborne | Taklamakan sieved | Tenerife airborne |
|---|---|---|---|---|---|---|---|
| Biotite | | | | | | | |
| Calcite | 67.2±1.2 | 81.0±1.0 | 11.0±0.5 | 6.0±0.3 | 33.0±0.6 | 14.6±0.4 | 6.6±0.3 |
| Chlorite | | | 7.5±1.4 | 5.9±1.0 | 2.7±9.5 | 7.1 ±0.9 | 1.6±0.6 |
| Cristobalite | | | | | | | |
| Dolomite/Ankerite | 8.0±0.4 | 1.3±0.2 | 9.0±0.5 | 1.4±0.6 | 4.6±0.5 | 4.6±0.5 | 2.2±0.3 |
| Gypsum | 1.2±0.2 | | | | | 1.8±0.5 | 2.4±0.3 |
| Halite | | | | | | 0.9±0.2 | 0.4±0.1 |
| Hematite | 0.5±0.1 | | 0.7±0.2 | 2.7±0.5 | 0.6±0.2 | | 0.6±0.1 |
| Hornblende | | | 0.6±0.3 | | 1.5±0.5 | 5.4±0.5 | |
| Illite | 4.2±1.6 | 0.3±0.2 | 8.0±1.7 | 3.0±0.7 | 12.5±1.0 | | 6.2±0.8 |
| Kaolinite | 0.8±0.6 | 0.3±0.3 | | | 7.8±0.7 | | 15.6±1.0 |
| Microcline | 1.7±0.4 | 1.3±0.4 | | 3.8±1.0 | | | 3.9±0.5 |
| Muscovite | | 1.1±0.5 | 9.6±0.7 | 1.8±0.3 | 4.8±0.8 | 8.0±0.6 | 7.4±0.6 |
| Orthoclase | | | 4.8±0.4 | 2.2±0.6 | 4.0±0.5 | 5.3±0.6 | |
| Palygorskite | 2.2±0.4 | 1.6±0.4 | | | 5.3±0.8 | | 4.1±0.4 |
| Na-plagioclase | 2.3±0.5 | 0.7±0.3 | 10.0±0.8 | 8.8±0.4 | 5.0±0.4 | 19.3±0.8 | 3.5±0.4 |
| Smectite | 4.5±0.6 | 6.4±1.2 | 26.1±2.6 | | | | 31.8±1.5 |
| Quartz | 7.3±0.3 | 6.1±0.2 | 12.8±0.5 | 63.8±1.2 | 17.9±0.4 | 33.1±0.7 | 14±0.4 |
| others | | | 0.2±0.1 | | 0.5±0.1 | | |

**Table 4.** Overview of the dust $n_s$ fit parameters $a$ and $b$, the resulting $R^2$ and the number of data points in each fit, $N$.

| Sample number | collection site | type | $a$ $(\mathrm{K}^{-1})$ | $b$ | $R^2$ | $N$ |
|---|---|---|---|---|---|---|
| 1 | Atacama | sieved | 0.513 | 9.39 | 0.91 | 36 |
| 2 | Atacama | milled | 0.363 | 14.50 | 0.96 | 50 |
| 3 | Australia | milled | 0.274 | 18.93 | 0.89 | 16 |
| 4 | Crete | airborne | 0.545 | 7.32 | 0.98 | 30 |
| 5 | Dubai | sieved | 0.391 | 13.04 | 0.96 | 35 |
| 6 | Egypt | airborne | 0.390 | 13.22 | 0.96 | 41 |
| 7 | Etosha | sieved | 0.289 | 17.09 | 0.93 | 33 |
| 8 | Great Basin | sieved | 0.286 | 15.47 | 0.93 | 35 |
| 9 | Israel | sieved | 0.477 | 10.11 | 0.91 | 39 |
| 10 | Israel | milled | 0.777 | -1.43 | 0.95 | 13 |
| 11 | Mojave | sieved | 0.317 | 15.62 | 0.96 | 46 |
| 12 | Morocco | milled | 0.257 | 18.55 | 0.95 | 45 |
| 13 | Peloponnese | airborne | 0.535 | 6.84 | 0.95 | 17 |
| 14 | Taklamakan | sieved | 0.355 | 15.00 | 0.93 | 21 |
| 15 | Tenerife | airborne | 0.455 | 10.16 | 0.97 | 14 |

**Table 5.** Overview of the Pearson correlation coefficients of the sum of selected mineral fractions and $\ln(n_s)$ at different temperatures. The names of the included sample numbers can be found in Tables 1 and 4. At 253 K only four samples were measured. At $T \leq 245$ K only samples with $FF$ between 0.1 and 0.9 were included for the correlations. The Etosha sample (7) was only included in the correlation with calcite because it does not contain feldspars, illite and kaolinite and only traces of quartz. An asterix indicates that the correlation was significant at the 0.05 level.

| $\ln(n_s)$ at | 253 K | 245 K | 243 K | 240 K | 238 K |
|---|---|---|---|---|---|
| number of samples | 3 (4) | 7 | 11 (12) | 13 (14) | 13 (14) |
| samples included | 2,6,(7),14 | 2,3,6,8,11,12,14 | 1,2,3,5,6,(7),8,9,11,12,13,14 | 1,2,4,5,6,(7),8,9,10,11,12,13,14,15 | |
| K-feldspar | 0.97 | -0.42 | -0.13 | 0.05 | -0.14 |
| feldspars | 0.87 | -0.29 | -0.12 | 0.37 | 0.30 |
| quartz | -0.96 | 0.89* | 0.91* | 0.52* | 0.45 |
| illite | | -0.11 | -0.39 | -0.14 | -0.16 |
| kaolinite | -0.35 | -0.66 | -0.47 | -0.55 | -0.61* |
| feldspars + quartz | 0.66 | 0.78* | 0.77* | 0.65* | 0.54 |
| feldspars + quartz+ illite | 0.66 | 0.80* | 0.73* | 0.62* | 0.51 |
| feldspars+quartz+kaolinite | 0.74 | 0.65 | 0.69* | 0.50 | 0.37 |
| feldspars+quartz+illite+kaolinite | 0.74 | 0.68 | 0.64* | 0.47 | 0.34 |
| calcite | -0.81 | -0.58 | -0.49 | -0.45 | -0.36 |

**Table 6.** Overview of the Pearson correlation coefficients of the sum of ice-active minerals and $\ln(n_s)$ at different temperatures of the samples where the mineralogy is representative for the size fraction smaller than 2.5 $\mu$m. The names of the included sample numbers can be found in Tables 1 and 4. Only samples with $FF$ between 0.1 and 0.9 were included for the correlations. The Etosha sample (7) was only included in the correlation with calcite because it does not contain feldspars, illite and kaolinite and only traces of quartz. An asterix indicates if the correlation was significant at the 0.05 level.

| $\ln(n_s)$ at | 243 K | 240 K | 238 K |
|---|---|---|---|
| number of samples | 4 (5) | 6 (7) | 6 (7) |
| samples included | 1,2,3,(7),13 | 1,2,4,(7),10,13,15 | |
| K-feldspar | 0.04 | 0.89* | 0.79 |
| all feldspars | -0.30 | 0.95* | 0.88* |
| quartz | 0.91 | 0.09 | 0.12 |
| illite | -0.80 | -0.05 | -0.17 |
| kaolinite | -0.56 | -0.40 | -0.36 |
| feldspars + quartz | 0.87 | 0.97* | 0.90* |
| feldspars + quartz+ illite | 0.84 | 0.93* | 0.84* |
| feldspars+quartz+kaolinite | 0.90 | 0.95* | 0.88* |
| feldspars+quartz+illite+kaolinite | 0.87 | 0.90* | 0.82* |
| calcite | -0.45 | -0.64 | -0.61 |