# Peer review of "Heterogeneous ice nucleation on dust particles sourced from 9 deserts worldwide - Part 1: Immersion freezing"

_Atmospheric Chemistry and Physics, 2016_

## Short Comment (SC1) · 17 Jun 2016

Mineral aerosol particles in the atmosphere and on the surface of soils rarely, if ever, do not contain associated biological material. As is well known, biological materials are the warmest IN observed in nature. Desert dusts are most readily generated in playas that are deposited by water that has concentrated small mineral particles and associated biological material into lower level areas. Desert dusts that travel appreciable distances rarely are sourced from the tops of sand (mineral) dunes. In this paper, it may be disingenuous to ascribe the measured IN content solely on the mineral content of the particles. The authors may consider heating the desert dusts in a furnace and boiling samples before and after the immersion IN freezing tests to remove the organics. If the

sample IN contents are unchanged, the authors and their audience may be somewhat assured that the mineral components of the samples are the IN source and not organic passengers. Russ Schnell, NOAA, Boulder, CO
* * *

---

## Referee Comment (RC1) · T. Zolles (Referee) · 8 Jul 2016

The paper gives new insights in the influence of natural dusts on ice nucleation. It is the first study relating the ice nucleation to the mineral composition of the natural dust samples. The authors very well extend recent studies on the ice nucleation of mineral dusts. Natural dusts are often rather complex mixtures of different minerals and single minerals are rare. This study addresses this by analyzing the composition of the dusts as well as discussing the influences of ground and airborne sampling. Some samples were milled which may alter the mineralogical composition. The authors are well aware of the problems of the bulk analyzing techniques and address the inhomogeneity of the mineralogy of dust with particle size. The study uses well established measurement

techniques. The study is able to show that findings from single minerals are in accordance with natural dust. It was possible to show that the ice nucleation ability of natural dusts can be related directly to their mineralogical composition.

**Sampling and sample treatment** The decision to sample and compare air-born and ground samples is very well justified. As was stated the mineralogical composition between the used <2.5 $\mu$ m fraction and the bulk (<32) that was analyzed with XRD may vary, but the general findings are very likely not influenced as neither mineral will be totally absent. This may influence the correlations in 3.3. Nevertheless, natural dusts and minerals are rather often found to have organic and biological material absorbed on their surface. It cannot be excluded that some are left on the dust particle surface. Gently heat treatment could have been used to destroy all organic material. Secondly the dusts did undergo different treatments to produce sufficiently small grains. The surface samples were sieved or milled and with the Israel and Atacama sample two fractions were created. It is clear that the Australia and Morocco samples had to be milled due to not small enough dust fraction, but the authors give no reason why not of all other samples a milled and a sieved fraction was created. There is no clear explanation why in particular the Atacama and Israel samples were chosen. To study the effect of milling a few samples are probably sufficient, but this may lead to an increased uncertainty in the correlations in 3.3. If certain sample sites have an increased importance due to a milled and a sieved fraction.

**Mineralogy analysis** The authors give a good description of the shortcomings of the analysis method as well as that the composition may not be valid for the particle sizes used in the freezing experiment.

Page 4/30: Similarly, the milling of the Israel sample likely interfered with the preferred direction of the minor components in the sieved samples, leading to an observed reduction of these mineral fractions (e.g. illite, kaolinite, plagioclase) in the milled compared to the sieved sample

There is a reduction in every relative fraction apart from the very soft calcite which, as the authors state, is probably increasing due to its softness upon milling. Why are these minor components of particular interest? What is the potential explanation of the observation?

Page 7/14: Natural mechanical weathering thus likely has enhanced the clay mineral and calcite content in the 15 smaller particle fraction whereas feldspars and quartz tend to be found in the larger size fraction.

Is the same expected for the sieved and milled samples in this study? In respect to the above mentioned part, it would be important to conduct milling on all ground samples.

Page 7/20: Hence, the Atacama and Israel milled, Australia, Crete, Peloponnese and Tenerife samples consisted mainly of particles smaller than 2.5 $\mu$m and the mineralogy is representative for the particles on which ice nucleation was studied.

I suggest adding the word milled to Australia here also: Hence Atacama, Israel and Australia milled as well as the airborne samples . . . to clarify and emphasize the origins.

**Immersion freezing experiments and data treatment**

**Dust size distribution**

The authors found a bimodal fit for all samples. As a none-expert in the field of particle shapes figure 3 seems to me that this could be a result of the two used measurement techniques? Is there an explanation for the two modes based on hardness or mineral composition?

**Ice nucleation of desert dust**

The whole section is very well written and the authors highlight new results.

Page 10/28: As a consequence of the large mean surface area of the Great Basin sample its ns is shifted to the lowest values.

[Figure]

A shift can be misunderstood, maybe just: "shows the lowest ns values. "

**Role of mineralogy**

The first part is very well written and points out the difficulties with single minerals in case of micas, muscovite and ankerite. The authors again emphasis well that surface collected samples may not be representative for the atmosphere. In the following paragraphs the study tries to correlate ns with the mineral fractions. The aim of this is clear and the general findings may still hold true, but the statistical significance of the reported correlations is questionable. Firstly, the Pearson correlation coefficients (R ) should not be used as the only statistical measure indicating the correlation. It is furthermore not clear from the text and table 5 which minerals were used at which temperature for obtaining the correlation coefficient. This information should be added to the manuscript. Including more statistical measures is probably out of scope of this study, but the regressions and plots yielding to the correlations should be added to the supplement.

In respect to the content I find the term anti-correlation of ns for illite and other clays rather confusing, independent of the fact that the values of |R|<0.5 have a low significance. Anti-correlation could be understood as an anti-freezing behavior. By increasing the content of example illite you reduce the ns at the given temperature, but in fact this is depending on the other mineral components.

The study reports that at 253K there is a good correlation between ns and the K-feldspar content, while 245K the correlation with quartz alone is better than with feldspars and quartz. The question arising is: how much of the total ns is still available at 245K to be explained by quartz if it had been already partly by K-feldspar at the higher temperature? By including a different amount of samples in the correlations at every temperature the relations are harder to identify. In my perspective it is necessary to also have a look at the correlations for the same 3 samples that were used at 253K alone at 245K and if they are better explained by quartz alone or joined by quartz and

feldspar. The authors do conduct a manual selection of which samples are included to obtain the correlation coefficient. I therefore want to emphasize once more to state which samples are taken for which correlation and to add this correlation plots to the supplement.

**Minor adjustments:** In figure 5 the great basin sample for me does not appear in the same color in the plot and the legend below 150

In table two what is the order of the minerals. It does not seem to be milled, sieved or airborne neither are they alphabetically sorted. Do you want to have the Israel samples on the same page?

---

## Referee Comment (RC2) · Anonymous Referee #2 · 10 Jul 2016

Review of "Heterogeneous ice nucleation on dust particles sourced from 9 deserts worldwide - Part 1: Immersion freezing" by *Boose et al.*

**Summary**
The authors investigated the relationship between the immersion freezing behavior of diverse natural dusts from nine desert regions around the globe (4 airborne and 11 sieved/milled surface samples) and their mineralogical compositions (that are based on bulk XRD). The immersion freezing measurements were conducted in the wide range of sub-zero temperatures (235-263 K) using two independent ice nucleation (IN) measurement techniques (i.e., mainly by IMCA-ZINC-IODE for $T$ below ~250 K and partially by FRIDGE for selected samples at temperatures above ~250 K). The authors evaluated the immersion freezing behavior in two metrics, frozen fraction (*FF*) and ice active surface-site density, $n_s(T)$.

The authors suggest that their dust samples exhibit somewhat different freezing behavior potentially due to variations in mineralogy (quartz and K-feldspar seem highly efficient immersion freezing components of dust) and sampling preparation procedures prior to the aerosolization (e.g., sieving and milling). More interestingly, the authors suggest that the surface-collected samples contain more efficient INPs, inferred by *FF* and $n_s(T)$, as compared to the airborne samples. The authors imply that mineralogy may play a significant role to explain the observed difference (P12 L18-20; P12 L25-27). Yet, they point out that further rigorous composition-resolved (i.e., both bio- and mineral-resolved) IN study of airborne dusts are necessary (P16 L8-13).

**General comments**
The experimental approach presented in this work is rigorous, and the authors undoubtedly put an enormous amount of effort to examine a variety of samples with careful sample preparation methods. However, the authors need quite a bit more work to improve the clarity regarding their analysis as the current manuscript contains some misleading statements and over-interpreted results. Critical comments are listed below. Six major comments are listed first, followed by the section-based specific and technical revisions. Additionally, I encourage the authors of the manuscript to thoroughly proof-read and language-check their manuscript as the list of specific/technical comments is too long.

Having said that, the research questions raised in this study (i.e., airborne dust vs. surface dust, mineralogy-resolved IN study) are interesting, and the research topic is certainly an important addition to ACP. If the authors improve the clarity of the paper by addressing the comments/suggestions below, the manuscript would be suitable for publication in ACP.

**Major comments**
1. *Kaufmann et al.* vs. *Boose et al.*: One of the major conclusions from this study (that is, mineralogy matters for dust IN) goes in the opposite direction of that from the study done by *Kaufmann et al.* (2016, AMTD; K16 hereafter), which points out the similar freezing behavior amongst multiple surface dust samples despite the difference/variation in mineralogy. Does this difference appear due to the technical issue/limitation (i.e., IMCA-ZINC-IODE vs. DSC) or totally something different? I am missing such discussion in this manuscript. Please elaborate to the extent of the community-wide scale. Providing a solid answer with respect to this point would be potentially important for modelers to consider if they require a computationally expensive mineralogy-resolved parameterization for their future simulation works or not.

Along similar lines, K16 suggests that microcline-containing particles are not abundant in the atmosphere and, hence, may have overall small contributions to atmospheric ice nucleation (IN) and glaciation (e.g., P20 L22-25 of K16). Do the authors agree with that statement? The reviewer thinks that microcline is in general an important dust composition that can contribute to efficient IN of material composites. Anyhow, it is somehow puzzling for the reviewer to see that two papers from the same institute report opposite things…

2. $n_{s, geo}$ vs. $n_{s, BET}$: As the authors may be aware, there are two sub-metrics in $n_s$, namely geometric-based $n_s$ and BET based $n_s$ (*Hiranuma et al.*, 2015, ACP; H15 hereafter). It appears to me that both $n_s$ metrics co-exist in Fig. 5 - Atokinson, Murray and Broadley seem BET-based, while Niemand and all of your data are geometric-based. This is an apples-to-oranges comparison. Note that the difference in these two $n_s$ metrics could be more than an order magnitude (e.g., for illite NX in H15), which would have a substantial effect on

your stats presented in Table 5 & P13 L5-33. I strongly urge the authors to select either one metric (the conversion method is introduced in H15), re-evaluate your stats and re-visit your statement in P1 L21-23. Your conclusion may change.

3. Your $n_s$ vs. Niemand's $n_s$: The parameterization described in *Niemand et al.* (2012; N12 hereafter) is essentially based on the observed ice number normalized to the measured "total" surface area (see Eqn. 1-4 in N12), while your parameterization introduces another approximated parameter, the weighted mean aerosol surface area ($A_{ve,w}$; P6 L11). Are these two $n_s$ values apples-to-apples? The performance comparison of IMCA-ZINC-IODE to the AIDA chamber, where the N12 study was conducted, with a reference IN material seems available in H15. The authors may extend the discussion given in H15 a bit more to justify the use of $A_{ve,w}$ in its first appearance.

In addition, the general discussion of the $n_s$ concept seems scattered over the multiple sections and somehow cumbersome (i.e., P11 L3-6, P9 L 4-7 and P10 L12-16). For clarity, I suggest the authors to briefly describe the following two things within one section (any):

1) Three assumptions of the $n_s$ parameterization, which is relevant to the immersion freezing characterization, include i) the probability of ice nucleation is proportional to the available surface area of immersed aerosols, ii) ice nucleation active sites are uniformly distributed over individual particle surfaces and iii) ice nucleation occurs at specific site in a deterministic manner (predominantly $T$ dependent).

2) The use of the weighted mean aerosol surface area ($A_{ve,w}$) is something unique in this study (and different from N12).

Accordingly, I suggest revising the conclusion (P 16 L15-17).

4. The effect of milling: The discussion regarding the effect of milling is misleading. Specifically, the reviewer's concerns are as follows:

P1 L18-20: This statement is misleading as it sounds like the milling process generally deteriorates IN efficiency of any composite materials. In fact, this statement contradicts to the IN results of the Atacama samples ($n_{s,milled} > n_{s,sieved}$) presented in Sect. 3.4. (i.e., P14 L29-30). I suggest rephrasing this sentence to be more specific (adding the "may" word would not help).

P15 L11-13: Without any evidence of the alternation in defect densities on the Islalel dust surface or calcite surface per milling, this statement (morphology vs. mineralogy) seems too ambitious/strong. It is likely the micro pores on the surface (*Subramanyam et al.*, 2016, Appl. Mater. Interfaces, doi:10.1021/acsami.6b01133) that enhance the IN activity of particles due to the inverse Kelvin effect (*Marcolli et al.*, 2014, ACP). If calcite etc. is breaking up but maintaining smooth surface, there is no reason for its fragments to enhance IN activity. The authors may be aware, but the nature of active sites is still uncertain and under investigation. Concerning these points, I suggest the authors to soften the tone of this statement.

5. Airborne vs. Surface: To me, the essence of this paper is summarized in P10 L9-10. This single statement and the method to reach this point would be worth a paper. The authors have already provided a great conclusion (P16 L8-10 & P1 L9-10). Focus your story along with this line. I have some questions regarding airborne dust vs. surface-collected dust as follows:

P11 L9-10: Interesting. The authors are right - according to Table 4, airborne samples in general seem having larger $a$ (0.48 ± 0.07 $K^{-1}$) as compared to the average of the rest (0.38 ± 0.16 $K^{-1}$), suggesting high $T$ dependency of the airborne samples. But, this is not all about $T$ dependency. This may be rather an indication that the airborne particles miss certain active sites that can be activated at high $T$ (owing to the difference in mineralogy???). The authors may discuss and clarify this point. This observation seems important.

P12 L26-27: This is another important statement. Elaborate a bit further by discussing the atmospheric relevancy of the minerals uniquely found in the airborne samples. Put clear emphasis if your bulk measurements at least suggest the atmospheric relevance of quartz/K-feldspar. According to P7 L14-15 and P7 L20-22, I have a feeling that there would be reduced amount of quartz/K-feldspar in air. Any comments? This argument seems the core research question of this manuscript.

Table 3: It is really bothering me that the Tenerife sample (6% microcline!) is not showing any superb IN behavior as compared to other airborne dusts. The authors said that the mineralogy inferred by XRD is representative of the aerosolized Tenerife sample (P7 L28-30). The authors disregard the contribution of atmospheric processes (P14 L17-19). Then, what is limiting the IN of this particular dust?

6. Table 5 (P13 L5-P14 L24): This part includes a number of flaws (i.e., $n_{s, geo}$ vs. $n_{s, BET}$) and needs substantial improvements. In fact, with given limitations/assumptions in P14 L6-24, I am not convinced that these 'relative' correlations add much meaning to the manuscript. Do the authors really require this part to draw their conclusion? This whole statistics part of the manuscript could be deleted?

   Again, I generally agree that quartz and microcline are IN active and may have potential importance in the atmospheric IN. What would be more valuable to see is if there is any 'absolute' relation between mineralogy and IN. For example, the authors may explore if the natural dust $n_s$ (or $FF$) can be optimized/predicted by its composition and associated individual $n_s$ scaled to the surface fraction ($SF$) of each component ($x_1, x_2$ … to $x_i$) [i.e., $n_{s,dust} = (n_{s,x1} \times SF_{x1}) + (n_{s,x2} \times SF_{x2}) + … + (n_{s,xi} \times SF_{xi})$]. Note that H15 attempted, but no success. Give more in-depth thoughts regarding the role of particular mineral IN.

**Specific comments/suggestions**
Introduction
P2 L22: Briefly explain what the authors mean for 'contradicting results'.
P2 L24: Briefly describe 'certain minerals'.
P2 L25: "For the implementation…" - I suggest starting a new paragraph here regarding the IN parameterizations. This way, the previous paragraph (L18-25) reads more like a general introduction of atmospheric INPs and their diversity (biological and non-biological).
P3 L19: The authors may mention that the abundance of quartz in atmospheric dusts is consistently high (i.e., ~10% in volume) in the size range of ~1 to 35 µm geometric diameter (see Table 1 of *Kandler et al*., 2009, Tellus; cited in this paper), which would add the atmospheric relevance and general importance of quartz. Such information could also fit in P4 L1-3.

Methods
P5 L26-27: Does the size distribution of particles in the chamber change over 3.5 hours (i.e., the filter sampling period; P8 L15-16)? Large particles settle down faster than the smaller ones, and the authors infer that certain mineral compositions are large in their sizes (e.g., P15 L33). Please clarify and discuss potential consequences, if any.
P5 L28: This background (~10%) seems high for the IN research. Any justification or measurements that this much background particles have no substantial contributions to heterogeneous freezing?
P5 L30: Briefly describe "coincidence effects" or cite proper papers.
P7 L2: According to Table 3 of K16 (*Kaufmann et al*.), the Israel sample contains some sanidine. Moreover, sanidine in the natural surface samples seems non-negligible as 6 out of 12 natural surface samples examined in K16 shows the presence of sanidine. Was your sample totally different from K16? Or the the sample may not be completely homogeneous in terms of mineralogical distribution even within a same batch? Was the XRD data interpretation method somewhat different from K16? Please clarify.
P7 L20-22: Doesn't this just mean that a majority of large particles (up to 32 µm) break up by the RBG milling? I mean that RBG may do more than just aerosol dispersion, correct? Long story short, is it really fair to assume that those aerosolized particles are identical to the sieved bulk used for XRD as the authors mention in P13 L20-23? Further clarification seems necessary. Accordingly, the authors may consider rephrasing the relevant text in the conclusion (i.e., P15 L29-32). 2.5 µm sounds like a magic number as it is right now.
P7 L28-30: For the reason given above, I am not sure if the authors can asset like this.

Results & Discussion

P10 L3: The homogeneous freezing regime presented in this manuscript is based on CNT? Or anything different? I suggest adding proper reference(s) here at least.

P10 L9-10: I encourage the authors to clearly state that the same trend holds true for another metric, $n_s(T)$.

P11 L2-3: It looks to me that the Niemand parameterization falls in the middle of your 15 $n_s(T)$ spectra. Please clarify what "rather at the lower end" means quantitatively.

P11 L13-14: The word "overpredicts" implies that the Niemand parameterization (N12) is wrong. The authors may want to soften the tone and rephrase. Your assumption vs. assumption made in N12 would be discussed here.

P11 L11-20: I do not find the scientific significance of having FRIDGE data included in this manuscript. The authors briefly discuss about the FRDIGE results in this part and only a bit more afterward. The reviewer does not find that the FRIDGE results are complementing IMCA-ZINC-IODE, vice versa. In addition, no proper justification for why FRIDGE was conducted for a subset of samples is provided. Does the authors' conclusion change without the FRIDGE data?

Fig. 5: Adding quartz reference spectrum (*Atkinson et al.*, 2013) would be nice.

Fig. 5 Cont'd: Having another panel depicting the highest-median-lowest spectra of airborne $n_s$ vs. those of surface-collected $n_s$ would add some clarity and strengthen the paper.

P12 L4-9: The discussion given here makes the review think something other than minerals (e.g., P5 L15-17) are competing for IN at given $T$ range. The reviewer is aware that the focus of the current work is on mineralogy vs. IN (and no biological INP perspective at all). That said, the authors should extend the discussion regarding the potential bioaccessibility of dust surface (*Augustin-Bauditz et al.*, 2016, ACP; *O'Sullivan et al.*, 2016, ACP) and other IN species that might be present in soil (*O'Sullivan et al.*, 2014, ACP; *Tobo et al.*, 2014, ACP; *Pummer et al.*, 2015, ACP). Such information would strengthen the manuscript.

Conclusion

P15 L18-19: Which data infers the *FF* and $n_s(T)$ results at temperatures above 250 K? I do not see them in Fig. 4. FRIDGE?

P16 L2-4: Your data presented in Tables 2 and 3 (i.e., Atacama milled vs. sieved; Israle milled vs. sieved) seem contradicting to this statement. For instance, I do not see any increase in the quartz fraction.

P16 L6-8: This part seems contradicting to the previous statement (that is, P15 L33).

**Technical comments/suggestions**

P1 L1-2: "Traditionally, clay minerals were assumed to determine…" → "Since natural dusts are composite in nature, clay minerals were typically used as a proxy to determine…"

P1 L14: "…activity in a given sample above 253 K that can be attributed to…"

P1 L17 and hereafter: Use the Italic font for *T* throughout the manuscript.

P2 L3: "determine" → "influence" - Besides primary ice nucleation, secondary ice processes can also contribute to the lifetime of clouds.

P2 L20: *Pummer et al.* missing as INM references

P2 L21: *DeMott et al.* missing as soot IN references

P2 L32: ", respectively"

P2 L32-P3 L1: Too many things packed in a single sentence. Split into two sentences.

P2 L33: between → amongst

P3 L13: "K-feldspars (microcline, orthoclase and sanidine) were…"

P3 L20-22: Redundant (P3 L4-6)

P3 L27: Delete "hence". Size and composition are inherently related. Is that what the authors want to say? If so, state so.

P3 L31: No "hence"

P4 L29: "The GPS coordinates of our collection sites…"

P5 L15-16: "The composition of all natural samples are presumably heterogeneous…"

P6 L10: "… sample (see …"

P6 L15: "… (sieving/milling)"

P8 L5: "The cloud droplets" → "The simulated cloud droplets" or "The activated droplets"
P8 L15: "subsequent" → "independent"
Eqn. 4: Missing negative sign on the RHS of Eqn.4?
P9 L4-5: Awkward sentence. I suggest rephrasing.
P9 L14-15: Awkward sentence. Simply say something like our measurements are valid with $\chi$ in the range of 1.1 to 1.6.
P9 L21-22: Awkward sentence. Simply say …because of the presence of predominantly large particles… or something similar.
P 10 L12-13: "Due to the heterogeneous and possibly size-dependent particle compositions, a partial step function like activation spectrum could be…"
P10 L27-28: two consecutive "overalls" are bothering.
P11 L2: "The comparison between the Niemand parameterization and our parameterizations based on…"
P11 L5: "polydisperse nature" → "heterogeneous properties"
P11 L23-24: This sentence does not fit in here. I suggest deleting.
P14 L34: The authors may want to remind the reader that 2.5 µm is $D_{50}$ of your cyclone.
P15 L10-13: "However… Thus…" - Awkward transition. Rephraze.
P15 L22 above or below?
P15 L27-28: I do not understand this sentence. Rephrase.
P15 L29-32: Awkward sentence. Rephrase.
P16 L 12: "can not" → "cannot" (to be consistent with the rest of the manuscript)
P16 L 21-22: Redundant (P4 L22)

---

## Author Comment (AC1) · 26 Sep 2016

We thank Russ Schnell for his interactive comment. We reproduce reviewer comments in blue in the following. Amended versions of the paper are given in *italics* for new sections and smaller red text for the original text. We have numbered the reviewer comments for clarity.

As major changes, we have

- renamed the "Namib sample" to "Etosha sample" throughout the manuscript to be more consistent with Kaufmann et al. 2016
- replaced "IN" with "ice nucleation" throughout the manuscript to avoid confusion with "ice nuclei"
- deleted p.9,l. 24-29 because the size distribution of the Australia sample was re-measured using an SMPS and APS. The mean surface area was very close to the original one (within 3%). The $n_s$ values in the revised manuscript include the updated surface area but have changed insignificantly
- binned the IMCA-ZINC $FF$ and $n_s$ data into 1 K bin for visual clarity in Fig. 4
- split up the original Table 5 into Tables 5 and 6 in the revised manuscript
- re-calculated Tables 5 (and 6) after we realized that it was incorrect to correlate the mineralogical fraction with $n_s$. Instead we correlated the fractions now to $\ln(n_s)$. The trends did not change but the $R$ values changed (typically by 0.03-0.1)
- added sample numbers in Tables 1 and 4 which we refer to in Tables 5 and 6
- added scatter plots (Figure 1 and 2) to the supplementary material corresponding to the correlation analysis in 3.3
- added Figure 3 to the supplementary material, showing the correlation of freezing temperatures with mineralogical fraction which were taken from Kaufmann et al. 2016

*Mineral aerosol particles in the atmosphere and on the surface of soils rarely, if ever, do not contain associated biological material. As is well known, biological materials are the warmest IN observed in nature. Desert dusts are most readily generated in playas that are deposited by water that has concentrated small mineral particles and associated biological material into lower level areas. Desert dusts that travel appreciable distances rarely are sourced from the tops of sand (mineral) dunes. In this paper, it may be disingenuous to ascribe the measured IN content solely on the mineral content of the particles. The authors may consider heating the desert dusts in a furnace and boiling samples before and after the immersion IN freezing tests to remove the organics. If the sample IN contents are unchanged, the authors and their audience may be somewhat assured that the mineral components of the samples are the IN source and not organic passengers. Russ Schnell, NOAA, Boulder, CO*

At the comparably low investigated temperatures (≤ 253 K), the role of biological particles is expected to be rather secondary (O'Sullivan et al. 2015) since some minerals are known to be efficient INP at these temperatures (e.g. Atkinson et al. 2013) and the concentration of dust particles compared to biological material is likely high (e.g. Fig. 19 of Murray et al. 2012). The focus of our study was therefore mineralogical composition of natural (airborne and surface-collected) dust samples and its role for ice nucleation. The small post-processing amount of most samples (particularly of the airborne ones) does

not allow a systematic study on the effect of heating of the samples on the ice nucleation properties with the same instrumentation (there is simply not enough dust left to repeat the measurements with heated samples). However, we were able to heat some of the samples at 300°C (573 K) for 10h and perform measurements in the condensation mode with the portable ice nucleation chamber, PINC (Chou et al. 2011), at a temperature of 240 K and 242 K (Fig. 1 below). We show here the results at $RH_w$ = 102-105 (240 K) and 100-103% (242 K), the highest measured $RH_w$. The measurements are not directly comparable to the immersion freezing results obtained with IMCA-ZINC since these $RH_w$ values do not guarantee complete droplet activation prior to freezing during the residence time of the particles in PINC (about 5 s). Nevertheless, they give an indication of the effect heating has on the ice nucleation ability of the particles.

[Figure]

**Figure 1: Condensation mode $n_s$ for selected heat treated samples.**

Figure 1 shows the condensation mode $n_s$ at 240 and 242 K. Filled symbols refer to unheated, open symbols to heated samples. Heating had little to no effect on the ice nucleation activity of the Australia, the Atacama milled and Peloponnese samples. The Etosha sample lost most of its ice nucleation activity after heating, which is the sample for which we could not relate the comparably high ice nucleation activity to its mineralogy as it consisted mostly of calcite and dolomite, which have a low ice nucleation activity (Atkinson et al. 2013, Kaufmann et al. 2016), and ankerite, of which the ice nucleation activity is not known (Kaufmann et al. 2016). This is consistent with the manuscript conclusions that the ice nucleation activity is mainly caused by the mineralogy of the dust samples. The situation may be different at warmer temperatures but unfortunately we cannot repeat investigations at these conditions due to the small remaining sample size. The full data from these additional measurements and further analysis will be shown in the second paper of the series (Boose et al. 2016).

We have added a paragraph on non-mineral matter mixed with the dust and its potential effects on the ice nucleation ability of the dust (p.4,l.15-20 of the revised manuscript):

*"Non-mineral matter, which can become internally or externally mixed with the mineral dust before or after emission, may affect the ice nucleating behavior of the dust. Sulfuric acid (Sullivan et al., 2010; Augustin-Bauditz et al., 2014) or secondary organic aerosol coating (Möhler et al., 2008a) has been*

*observed to decrease the ice nucleating ability while exposure to ozone (Kanji et al., 2013) or the presence of ammonium sulfate (Boose et al., 2016b) has been suggested to improve it. Biological material can adsorb to mineral dust, keeping its ice nucleating ability and lead to an increased ice-nucleating ability of the dust (Schnell, 1977; Conen et al., 2011; O'Sullivan et al., 2016)."*

We have rephrased p.5,.15-16: "All natural dust samples are expected to be very heterogeneous, i.e. external and internal mixtures of different minerals and potentially biological material (Meola et al., 2015)."

to now p.5,l.28-20 of the revised manuscript:
*"The composition of natural dust samples is presumed to be heterogeneous, i.e. external and internal mixtures of different minerals and potentially containing organic or biological material (Meola et al., 2015). "*

We have replaced p.14, l.22-23:

"If this is due to reasons other than mineralogy such as coating as suggested by Kaufmann et al. (2016) or if the present minerals ankerite, dolomite or muscovite can lead to a high IN activity at T < 243 K under certain circumstances is not known."

with now p.15,l.15-16 of the revised manuscript:

*"Measurements in the condensation mode, which are the subject of part 2 of this study, suggest that the ice nucleation activity of this sample is in large part related to organic or biological material mixed with the dust"*

We have replaced p.14, l.17-19:

"The results suggest that potential coating of the particles only plays a secondary role for the immersion freezing ability of the mineral dusts as the majority of the results can be explained by the mineralogy as has also been observed by Kaufmann et al. (2016)."

with now p.15, l.10-13 of the revised manuscript:

*"While the correlations do not exclude an influence of non-mineral material, the results suggest that potential coatings or mixing of the particles only play a secondary role for the immersion freezing ability at the studied temperatures of the mineral dusts. The majority of the results can be explained by the mineralogy as also observed by Kaufmann et al. (2016). "*

**References:**

Boose, Y., et al.: Heterogeneous ice nucleation on dust particles sourced from 9 deserts worldwide - Part 2: Deposition and condensation freezing, in prep., 2016

Chou, C., Stetzer, O., Weingartner, E., Jurányi, Z., Kanji, Z. A., and Lohmann, U.: Ice nuclei properties within a Saharan dust event at the Jungfraujoch in the Swiss Alps, Atmos. Chem. Phys., 11, 4725–4738, doi:10.5194/acp-11-4725-2011, 2011.

Murray, B. J., O'Sullivan, D., Atkinson, J. D., and Webb, M. E.: Ice nucleation by particles immersed in supercooled cloud droplets, Chem. Soc. Rev., 41, 6519, doi:10.1039/c2cs35200a, 2012.

O'Sullivan, D., Murray, B. J., Ross, J. F., Whale, T. F., Price, H. C., Atkinson, J. D., Umo, N. S. and Webb, M.E.: The relevance of nanoscale biological fragments for ice nucleation in clouds, Sci. Rep. 5, 8082, doi: 10.1038/srep08082, 2015.

---

## Author Comment (AC2) · 26 Sep 2016

We thank Reviewer T. Zolles for his constructive comments. We reproduce reviewer comments in blue in the following. Amended versions of the paper are given in *italics* for new sections and smaller red text for the original text. We have numbered the reviewer comments for clarity.

As major changes, we have

- renamed the "Namib sample" to "Etosha sample" throughout the manuscript to be more consistent with Kaufmann et al. 2016
- replaced "IN" with "ice nucleation" throughout the manuscript to avoid confusion with "ice nuclei"
- deleted p.9,l. 24-29 because the size distribution of the Australia sample was re-measured using an SMPS and APS. The mean surface area was very close to the original one (within 3%). The $n_s$ values in the revised manuscript include the updated surface area but have changed insignificantly
- binned the IMCA-ZINC *FF* and $n_s$ data into 1 K bin for visual clarity in Fig. 4
- split up the original Table 5 into Tables 5 and 6 in the revised manuscript
- re-calculated Tables 5 (and 6) after we realized that it was incorrect to correlate the mineralogical fraction with $n_s$. Instead we correlated the fractions now to $\ln(n_s)$. The trends did not change but the $R$ values changed (typically by 0.03-0.1).
- added sample numbers in Tables 1 and 4 which we refer to in Tables 5 and 6
- added scatter plots (Figure 1 and 2) to the supplementary material corresponding to the correlation analysis in 3.3
- added Figure 3 to the supplementary material, showing the correlation of freezing temperatures with mineralogical fraction which were taken from Kaufmann et al. 2016

**1) Sampling and sample treatment** *The decision to sample and compare air-born and ground samples is very well justified. As was stated the mineralogical composition between the used <2.5 μm fraction and the bulk (<32) that was analyzed with XRD may vary, but the general findings are very likely not influenced as neither mineral will be totally absent. This may influence the correlations in 3.3. Nevertheless, natural dusts and minerals are rather often found to have organic and biological material absorbed on their surface. It cannot be excluded that some are left on the dust particle surface. Gently heat treatment could have been used to destroy all organic material.*

We agree with the reviewer's comment (see also response to SC by Russ Schnell). We had performed parallel measurements of particle fluorescence as a proxy of biological material, but due to a failure of the xenon lamps these data were not useable. Unfortunately due to the small sample amounts, we are limited in repeating or performing additional measurements. We were able to repeat some of the deposition/condensation measurements with heat treatment as suggested by Reviewer Zolles. We will include the results of these in the part 2 paper associated with this manuscript (Boose et al., 2016). The experiments were performed using the portable ice nucleation chamber PINC with the original samples as well as samples heated at 300°C (573 K) for 10h. PINC, which works like ZINC but is shorter, has a shorter residence time and sampled particles which were dry rather than pre-activated as droplets as

was done in the present work. Therefore, the results are not necessarily comparable to the immersion mode results from the current manuscript because full droplet activation prior to freezing is not guaranteed.

Figure 1 shows the condensation mode $n_s$ at 240 and 242 K. Filled symbols refer to unheated, open symbols to heated samples. Heating had little to no effect on the ice nucleation activity of the Australia, the Atacama milled and Peloponnese samples. Only the Etosha sample lost most of its ice nucleation activity after heating, which is the sample for which we could not relate the comparably high ice nucleation activity to its mineralogy as it consisted mostly of minerals with low ice nucleation activity. This is consistent with the manuscript conclusions that the ice nucleation activity is mainly caused by the mineralogy of the dust samples. The situation may be different at the warmer temperatures in the manuscript (> 250 K) but unfortunately we cannot do investigations at these conditions due to the small remaining sample size. The full data from these additional measurements and further analysis will be shown in the second paper of the series (Boose et al. 2016).

[Figure]

Figure 1: Condensation mode $n_s$ for selected heat treated samples.

In the current manuscript we have added a paragraph on non-mineral matter mixed with the dust and its potential effects on the ice nucleation ability of the dust (p.4,l.15-20 of the revised manuscript):

*"Non-mineral matter, which can become internally or externally mixed with the mineral dust before or after emission, may affect the ice nucleating behavior of the dust. Sulfuric acid (Sullivan et al., 2010; Augustin-Bauditz et al., 2014) or secondary organic aerosol coating (Möhler et al., 2008a) has been observed to decrease the ice nucleating ability while exposure to ozone (Kanji et al., 2013) or the presence of ammonium sulfate (Boose et al., 2016b) has been suggested to improve it. Biological material can adsorb to mineral dust, enhancing its ice nucleating ability (Schnell, 1977; Conen et al., 2011; O'Sullivan et al., 2016)."*

We have rephrased p.5,.15-16 (now p.5,l.28-30 of the revised manuscript):

"All natural dust samples are expected to be very heterogeneous, i.e. external and internal mixtures of different minerals and potentially biological material (Meola et al., 2015)."

*"The composition of natural dust samples is presumed to be heterogeneous, i.e. external and internal mixtures of different minerals and potentially containing organic or biological material (Meola et al., 2015). "*

We have replaced p.14, l.22-23:

"If this is due to reasons other than mineralogy such as coating as suggested by Kaufmann et al. (2016) or if the present minerals ankerite, dolomite or muscovite can lead to a high IN activity at T < 243 K under certain circumstances is not known."

with (now p.15,l.27-28 of the revised manuscript):

*"Measurements in the condensation mode, which are the subject of part 2 of this study, suggest that the ice nucleation activity of this sample is in large part related to organic or biological material mixed with the dust."*

*2) Secondly the dusts did undergo different treatments to produce sufficiently small grains. The surface samples were sieved or milled and with the Israel and Atacama sample two fractions were created. It is clear that the Australia and Morocco samples had to be milled due to not small enough dust fraction, but the authors give no reason why not of all other samples a milled and a sieved fraction was created. There is no clear explanation why in particular the Atacama and Israel samples were chosen. To study the effect of milling a few samples are probably sufficient, but this may lead to an increased uncertainty in the correlations in 3.3. If certain sample sites have an increased importance due to a milled and a sieved fraction.*

We agree that it would be preferable if all ground-collected samples could have been treated the same way. Unfortunately, this was not possible since the volume – fraction of particles < 32 µm was typically only on the order of 1/1000 or less of the initial, pre-preparation sample. Only the Israel sample had a large enough mass fraction of particles smaller than 32 µm to allow the direct comparison of a milled sieved fraction to the unmilled sieved fraction. The Atacama sample also had a small fraction of < 32 µm but enough to do a comparison between a milled non-sieved fraction and an unmilled sieved fraction. For none of the other samples the amount of particles < 32 µm was large enough to allow further comparisons.

We have added on p.5,l.27-28 of the revised manuscript:
*"The sub-32 µm size fraction of the other samples was too small to investigate the milling effect."*

We have repeated the correlations in section 3.3. including only either the sieved or milled Atacama and Israel samples. The trends stayed consistent with those presented in section 3.3. Therefore, an overrepresentation of these two sampling sites does not affect the conclusions presented in the paper.

*3) **Mineralogy analysis** The authors give a good description of the shortcomings of the analysis method as well as that the composition may not be valid for the particle sizes used in the freezing experiment.*
*Page 4/30: Similarly, the milling of the Israel sample likely interfered with the preferred direction of the minor components in the sieved samples, leading to an observed reduction of these mineral fractions (e.g. illite, kaolinite, plagioclase) in the milled compared to the sieved sample.*
*There is a reduction in every relative fraction apart from the very soft calcite which, as the authors state, is probably increasing due to its softness upon milling. Why are these minor components of particular interest? What is the potential explanation of the observation?*

For the ice nucleating behavior of the dust samples all components are of interest due to the unknown mixing state of the particles. In theory it could be that a minor component by weight is present on all particles and responsible for the ice nucleation. The minor components are mentioned here in regard to the preferred orientation during XRD to explain the differences between the Israel milled and sieved sample. In theory the mineralogy should be the same because the milled Israel sample was derived from milling the sieved Israel sample.

Preferred orientation of minerals can lead to an over- or underestimation of some minerals, which we suggest (p.7,l.10-14 of the revised manuscript) explains the difference between the milled and unmilled Israel sample.

*4) Page 7/14: Natural mechanical weathering thus likely has enhanced the clay mineral and calcite content in the smaller particle fraction whereas feldspars and quartz tend to be found in the larger size fraction. Is the same expected for the sieved and milled samples in this study? In respect to the above mentioned part, it would be important to conduct milling on all ground samples.*

The milling breaks up hard minerals such as quartz and feldspar and grinds them to smaller sizes. Therefore it changes the size distribution and size dependent mineralogy of the milled sample compared to the sieved sample. In this regards, the milled samples may show a different size dependent mineralogy compared to unprocessed dust. However, all unmilled samples, including the airborne ones, contain quartz and (except the Etosha (Namib) sample) also feldspar. Furthermore, as described on p.4,l.6-14 of the revised manuscript, quartz and feldspar are commonly found in airborne dust samples. Thus, the milled samples differ from the sieved samples but contain the same minerals found also in sieved or airborne samples.

Milling of all ground samples would allow a better overlap of particle sizes accessible with XRD and IMCA-ZINC. This would help answer one of our scientific questions, i.e. the correlation between mineralogical composition and ice nucleation activity. However, the studied samples would probably be less atmospherically relevant than the ones derived sieving the surface-collected samples.

*5) Page 7/20: Hence, the Atacama and Israel milled, Australia, Crete, Peloponnese and Tenerife samples consisted mainly of particles smaller than 2.5 μm and the mineralogy is representative for the particles on which ice nucleation was studied.*
*I suggest adding the word milled to Australia here also: Hence Atacama, Israel and Australia milled as well as the airborne samples . . . to clarify and emphasize the origins.*

We have deleted this sentence to avoid repetition. Instead, we rephrased the sentence on p.7,l. 28-30 of the original manuscript (now p. 8, l. 8-10):

"In summary, the identified mineralogical composition is well representative for the particle size fraction used for IN experiments on the Atacama milled and sieved, Israel milled, Australia, Crete, Peloponnese and Tenerife samples."

"*In summary, the identified mineralogical composition is well representative for the particle size fraction used for ice nucleation experiments on the Atacama milled and sieved, sieved Etosha, Israel milled, milled Australia, and the airborne Crete, Peloponnese and Tenerife samples.*"

*6) The authors found a bimodal fit for all samples. As a none-expert in the field of particle shapes figure 3 seems to me that this could be a result of the two used measurement techniques? Is there an explanation for the two modes based on hardness or mineral composition?*

We did not find a combination of shape factor and density that is realistic for dust particles which would lead to a size distribution of the dust samples with only one mode (p.9,l.14-15 of the original manuscript). It is reasonable that there is a very broad peak with two, sometimes even three shoulders related to the high inhomogeneity of the samples with respect to hardness and fracture. The majority of the mineral composition tends to be soft (e.g. calcite and clays), giving smaller particles, or hard (e.g. quartz and feldspar) resulting in larger particles.

We have added on p.10,l.2-3 (revised manuscript): "*They are likely related to the high inhomogeneity of the samples with respect to hardness and fracture.*"

*7) The whole section is very well written and the authors highlight new results.*
*Page 10/28: As a consequence of the large mean surface area of the Great Basin sample its ns is shifted to the lowest values.*
*A shift can be misunderstood, maybe just: "shows the lowest ns values. "*

Agreed. We have rephrased p.10,l.27-29:
"The Israel milled and the Peloponnese sample show a low IN activity. Overall, the airborne Saharan samples belong to the lower half of all ns curves. As a consequence of the large mean surface area of the Great Basin sample its ns is shifted to the lowest values."

now p.10,l.32-33 of the revised manuscript:
"*The $n_s$ of the Great Basin sample, which has one of the highest FFs, is amongst the lowest, due to its coarse particle sizes.*"

and p.11,l.6-7 of the revised manuscript:
"*The Israel milled, the Great Basin, and the Peloponnese sample show a low ice nucleation activity.*"

*8) Role of mineralogy*
*The first part is very well written and points out the difficulties with single minerals in case of micas, muscovite and ankerite. The authors again emphasis well that surface collected samples may not be*

*representative for the atmosphere. In the following paragraphs the study tries to correlate ns with the mineral fractions. The aim of this is clear and the general findings may still hold true, but the statistical significance of the reported correlations is questionable. Firstly, the Pearson correlation coefficients (R ) should not be used as the only statistical measure indicating the correlation. It is furthermore not clear from the text and table 5 which minerals were used at which temperature for obtaining the correlation coefficient. This information should be added to the manuscript. Including more statistical measures is probably out of scope of this study, but the regressions and plots yielding to the correlations should be added to the supplement.*

We have indicated now with an asterix which $R$ - values are significant at the 0.05 level for each correlation in Table 5 and the new Table 6. We have added the most important scatter plots yielding the correlations as Figure 1 to the supplement. These are the correlations of $n_s$ at 243 K, 245 K and 253 K with quartz, calcite, K-feldspars and kaolinite, respectively.
Furthermore, we have added the following lines at p.14,l.6-9 of the revised manuscript:

"*Table 5 shows the Pearson correlation coefficients (R) between the mineral fractions and the $n_s$ at five temperatures. Only a few of the correlations are statistically significant, owing to the low number of samples. Nevertheless, the overview of the correlation coefficients gives an idea of the effect of certain minerals on $n_s$. The related scatter plots are given in Figure 1 of the supplementary material.*"

To clarify which samples were used at which temperature we have added sample numbers in Tables 1 and 4 and provide these sample numbers in Table 5 and the new Table 6 as "samples included". We have split the original Table 5 in two: the new Table 5 contains the former first five columns and the new Table 6 the former last three columns. Furthermore, we clarify on p.14, l.2-5 of the revised manuscript:

"*For each temperature, all samples which showed a FF between 0.1 and 0.9 were used for the correlations. The Etosha sample was excluded from the correlations with feldspars, quartz and clays, as it does not contain any significant amount of these minerals. However, it was included for the comparison with calcite.*"

*9) In respect to the content I find the term anti-correlation of ns for illite and other clays rather confusing, independent of the fact that the values of |R|<0.5 have a low significance. Anti-correlation could be understood as an anti-freezing behavior. By increasing the content of example illite you reduce the ns at the given temperature, but in fact this is depending on the other mineral components.*

Rather than being relative to the fundamental process of freezing, the mineralogy correlation coefficients calculated are relative to the mean mineralogy of the samples. This means that if a sample has increased levels of (presumably) ice nucleation inactive minerals such as illite and calcite, it will have a correspondingly lower concentration of the (presumably) ice nucleation active quartz and feldspars. This produces the anticorrelation between these minerals and $n_s$. To avoid any confusion we have replaced "anticorrelation" with "*negative correlation*" and "anticorrelated" with "*negatively correlated*".

*10) The study reports that at 253K there is a good correlation between ns and the Kfeldspar content, while 245K the correlation with quartz alone is better than with feldspars and quartz. The question arising is: how much of the total ns is still available at 245K to be explained by quartz if it had been already partly by K-feldspar at the higher temperature? By including a different amount of samples in the correlations at every temperature the relations are harder to identify. In my perspective it is necessary to also have a look at the correlations for the same 3 samples that were used at 253K alone at 245K and if they are better explained by quartz alone or joined by quartz and feldspar. The authors do conduct a manual selection of which samples are included to obtain the correlation coefficient. I therefore want to emphasize once more to state which samples are taken for which correlation and to add this correlation plots to the supplement.*

In response to comment 8, we added sample numbers to Tables 1 and 4 and provide the sample numbers for each correlation (Table 5 and Table 6).

We abstained from presenting correlations at T < 253 K for only three samples due to the even lower significance. However, we have included them now as Figure 2 in the supplement. The correlations for only Taklamakan, Egypt, and Atacama milled at $T$ < 250 K are consistent with the correlations given for all relevant samples. Since none of the three samples contains any illite and only one sample contains kaolinite, the addition of clays to the correlation is meaningless. At 245 K the sum of quartz and all feldspar is the best predictor for $n_s$ ($R$ = 0.97) compared to the K-feldspars ($R$ = 0.68) or the sum of all feldspars ($R$ = 0.85).  At $T$=240 K and 243 K, the sum of quartz and feldspar correlates best with $n_s$. At T = 238 K finally, quartz alone leads to the best correlation ($R$ = 0.72) with $n_s$.

We have added on p.14, l.23-26 of the revised manuscript:

"*To exclude a bias from varying numbers of samples at different temperatures, we have repeated the correlations at 245, 243, 240 and 238 K for the Atacama milled, Egypt and Taklamakan samples only. The corresponding scatter plots are provided as Figure 2 of the supplementary material and confirm the observations that quartz plus feldspar yield the best correlations at T < 245 K.*"

*11) Minor adjustments: In figure 5 the great basin sample for me does not appear in the same color in the plot and the legend below 150*

Corrected.

*12) In table two what is the order of the minerals. It does not seem to be milled, sieved or airborne neither are they alphabetically sorted. Do you want to have the Israel samples on the same page?*

We have sorted the samples now alphabetically in all tables.

**References:**

Boose, Y., et al.: Heterogeneous ice nucleation on dust particles sourced from 9 deserts worldwide - Part 2: Deposition and condensation freezing, in prep., 2016

Chou, C., Stetzer, O., Weingartner, E., Jurányi, Z., Kanji, Z. A., and Lohmann, U.: Ice nuclei properties within a Saharan dust event at the Jungfraujoch in the Swiss Alps, Atmos. Chem. Phys., 11, 4725–4738, doi:10.5194/acp-11-4725-2011, 2011.

Kaufmann, L., Marcolli, C., Hofer, J., Pinti, V., Hoyle, C. R., and Peter, T.: Ice nucleation efficiency of natural dust samples in the immersion mode, Atmos. Chem. Phys., 16, 11177-11206, doi:10.5194/acp-16-11177-2016, 2016

---

## Author Comment (AC3) · 26 Sep 2016

We thank Reviewer 2 for their constructive comments. We reproduce reviewer comments in *blue* in the following. Amended versions of the paper are given in *green italics* for new sections and smaller red text for the original text.

As major changes, we have

- renamed the "Namib sample" to "Etosha sample" throughout the manuscript to be more consistent with Kaufmann et al. 2016
- replaced "IN" with "ice nucleation" throughout the manuscript to avoid confusion with "ice nuclei"
- deleted p.9,l. 24-29 because the size distribution of the Australia sample was re-measured using an SMPS and APS. The mean surface area was very close to the original one (within 3%). The $n_s$ values in the revised manuscript include the updated surface area but have changed insignificantly
- binned the IMCA-ZINC *FF* and $n_s$ data into 1 K bin for visual clarity in Fig. 4
- split up the original Table 5 into Tables 5 and 6 in the revised manuscript
- re-calculated Tables 5 (and 6) after we realized that it was incorrect to correlate the mineralogical fraction with $n_s$. Instead we correlated the fractions now to $\ln(n_s)$. The trends did not change but the *R* values changed (typically by 0.03-0.1)
- added sample numbers in Tables 1 and 4 which we refer to in Tables 5 and 6
- added scatter plots (Figure 1 and 2) to the supplementary material corresponding to the correlation analysis in section 3.3
- added Figure 3 to the supplementary material, showing the correlation of freezing temperatures with mineralogical fraction which were taken from Kaufmann et al. 2016

*1. Kaufmann et al. vs. Boose et al.: One of the major conclusions from this study (that is, mineralogy matters for dust IN) goes in the opposite direction of that from the study done by Kaufmann et al. (2016, AMTD; K16 hereafter), which points out the similar freezing behavior amongst multiple surface dust samples despite the difference/variation in mineralogy. Does this difference appear due to the technical issue/limitation (i.e., IMCA-ZINC-IODE vs. DSC) or totally something different? I am missing such discussion in this manuscript. Please elaborate to the extent of the community-wide scale. Providing a solid answer with respect to this point would be potentially important for modelers to consider if they require a computationally expensive mineralogy-resolved parameterization for their future simulation works or not.*

We do not agree with the reviewer's statement that the results of our study go in the opposite direction of the study by Kaufmann et al. (K16 hereafter) which is in ACP, not AMTD. On p. 11197 of K16, the authors clearly state: "*However, mineralogical composition does matter.*"

Furthermore, in the abstract of K16: "*In summary, the mineralogical composition can explain the observed freezing behaviour of 5 of the investigated 12 natural dust samples, and partly for 6 samples, leaving the freezing efficiency of only 1 sample not easily explained in terms of its mineral reference components.*"

However, the abstract of K16 also states:

"*All natural dusts, except for the Antarctica and ATD samples, froze in a remarkably narrow temperature range with the heterogeneously frozen fraction reaching 10% between 244 and 250 K, 25% between 242 and 246 K, and 50% between 239 and 244 K.*"

This statement is not in conflict with our results. The maximum temperature range found by K16 for the natural dust samples excluding the Antarctica and ATD samples is 6 K (frozen water fraction reaching 10 %) while the maximum $T$-range at the same $n_s$ in our data is 10 K. The larger spread is probably due to the larger number of dust samples and a greater variety in the dust mineralogy in our study, particularly in the quartz content (K16: 1-26 wt%, our study: 1-91 wt%), the K-feldspar content (K16: 0-10 wt%, our study: 0-30 wt%) and the sum of all feldspars (K16: 0-23 wt%, our study: 0-65 wt%).

In addition, the resolution of the latent heat release peaks in the differential scanning calorimetry (DSC) may result in masking of the variability in freezing. In the DSC a droplet can contain more than one particle while IMCA-ZINC works on a single particle basis. If several particles are contained in a droplet, the particle with the most active site will initiate freezing. K16 studied droplets of about 2 μm which contain only a few particles. This curtails the effects but could still cause some of the variability not being observed with the DSC compared to the IMCA-ZINC.

Both studies conclude that as a first approximation, a parameterization like the one by Niemand et al. 2012 is applicable but a more sophisticated parameterization should rely on the mineralogical composition of dust (p.11198 in K16 and p.17,l.21-23 in our revised manuscript).

To clarify this point we have rephrased the following sentence on p.3,l.27-33 of the revised manuscript:

"*A recent study by Kaufmann et al. (2016) investigated the ice nucleation ability of surface-collected samples from eight different arid regions worldwide and several single-mineral reference samples using differential scanning calorimetry. The authors found at maximum a 6 K spread in freezing temperatures of emulsion experiments amongst surface-collected samples from different atmospheric-dust source regions. They confirmed the exceptional freezing ability of microcline but found only a minor fraction (4 wt%) in one of the samples from the dust source regions studied. Their samples contained quartz fractions between 1 and 26 wt%, K-feldspar fractions between 0 and 10 wt%, and plagioclase fractions between 0 and 22 wt%.*".

Furthermore, we show in Figure 1 of this document correlation plots of $T_{het}$ at 10 and 50 % frozen water fraction of 0.5 and 1 wt% suspensions with the quartz, calcite, plagioclase, microcline and smectite fraction, all taken from K16. The highest correlation coefficient is found for microcline (R = 0.93, 0.85 and 0.93). The correlation with microcline however also suffers from a low number of samples. Microcline is followed by quartz (R =0.75, 0.72, and 0.79) and plagioclase (R = 0.53, 0.27, and 0.4) as best predictors for ice nucleation activity (here in terms of $T_{het}$ at 10 and 50 % frozen water fraction). These results show that our method works well and the conclusions are supported by K16.

[Figure]

**Figure 1: Correlation of freezing temperature of 10 and 50 % frozen water fraction of a 0.5 and 1 wt% solution with six different minerals for the data from Kaufmann et al. 2016**

*Along similar lines, K16 suggests that microcline-containing particles are not abundant in the atmosphere and, hence, may have overall small contributions to atmospheric ice nucleation (IN) and glaciation (e.g., P20 L22-25 of K16). Do the authors agree with that statement? The reviewer thinks that microcline is in general an important dust composition that can contribute to efficient IN of material composites. Anyhow, it is somehow puzzling for the reviewer to see that two papers from the same institute report opposite things…*

We already addressed this point in the original manuscript (p.12, l.10-12):

"We find microcline in at least three of the surface-collected (Great Basin and both Israel) and also in the airborne Tenerife sample. In the samples from Australia and Morocco both phases orthoclase and microcline seem to be present."

and on p.14, l. 6-10:

"We find microcline in some of our natural dust samples and also in one of the airborne samples. However, we cannot confirm a superior role of microcline over orthoclase for ice nucleation as in most samples microcline was found to be less than 8 wt%. The Great Basin sample contained 30 wt% microcline in the bulk sample but likely significantly less in the size fraction < 2.5 μm. The highly polymineral nature of our samples and the effect of differences in the size distributions prevent a further investigation of the role of microcline."

We have merged the two parts above in the revised manuscript (now p.13,l.10-13):

*"We find less than 8 wt% microcline in the sieved and milled Israel and the airborne Tenerife samples. In the samples from Australia and Morocco both K-feldspars orthoclase and microcline seem to be present. The surface-collected Great Basin sample contained 30 wt% microcline in the bulk sample but likely much less in the size fraction < 2.5 µm. "*

We have added a paragraph on the role of microcline in the conclusions (p.16,l.14-19 of the revised manuscript):

*"Microcline was found in one airborne sample (6 wt%) and in surface-collected samples from four different locations. We could not confirm a superior role of microcline over orthoclase, in-part because a differentiation of the two minerals was often difficult and partly because their content was too low in the size range investigated for ice nucleation to cause an effect detectable with the IMCA-ZINC experiment We cannot therefore conclude that microcline is generally atmospherically not relevant because even in low amounts of a few percent it could glaciate clouds at temperatures warmer than 253 K."*

We cannot confirm or negate a superior role of microcline over orthoclase due to the described limitations in the ice nucleation and mineralogy measurements in terms of concentration and temperature. We therefore abstain from any further speculation on the role of microcline in the atmosphere.

*2. ns, geo vs. ns, BET: As the authors may be aware, there are two sub-metrics in ns, namely geometric-based ns and BET based ns (Hiranuma et al., 2015, ACP; H15 hereafter). It appears to me that both ns metrics co-exist in Fig. 5 - Atokinson, Murray and Broadley seem BET-based, while Niemand and all of your data are geometric-based. This is an apples-to-oranges comparison. Note that the difference in these two ns metrics could be more than an order magnitude (e.g., for illite NX in H15), which would have a substantial effect on your stats presented in Table 5 & P13 L5-33. I strongly urge the authors to select either one metric (the conversion method is introduced in H15), re-evaluate your stats and re-visit your statement in P1 L21-23. Your conclusion may change.*

Figure 5 is a comparison between our results (all in $n_{s,geo}$) to literature curves to give an overview on the ice nucleation ability of mineralogical components. Furthermore, we do not use the literature values in our statistical analysis, so the conversion of the literature data to $n_{s,geo}$ doesn't have any effect on Table 5 (and now also Table 6) or the interpretation. Keeping this in mind, we agree with the reviewer, that a conversion of all $n_s$ curves into the same metric is a fairer comparison to the literature, despite the uncertainties inherent to such a conversion, particularly where values needed for such a conversion aren't reported. To acknowledge this uncertainty, we do not show single $n_{s,geo}$ curves in the revised Fig. 5 but instead the area between $n_{s,BET}$ and the calculated $n_{s,geo}$ curves.

To calculate $n_{s,geo}$ curves we have done the following:

-In the case of Atkinson et al. 2013 (A13 in the following), we have converted the K-feldspar $n_{s,BET}$ curve into $n_{s,geo}$ using a factor of 3.5 as given in the Supplementary Material of A13.

-In the case of Broadley et al. 2012 (B12 in the following), we have converted the illite $n_{s,BET}$ curve following Hiranuma et al. 2015, using the specific surface area of 104.2 m$^2$/g given in B12 and a ratio of

the total surface area to total mass of 6.54 m²/g given in Hiranuma et al. 2015 for the same illite NX sample.

- For the Murray et al. 2011 (M11 in the following) case, we converted the kaolinite $n_{s,BET}$ curve into $n_{s,geo}$, following Hiranuma et al. 2015, using a specific surface area of 11.8 m²/g (M11), a density of 2.63 g/cm³ (M11) and a mean mass- weighted diameter of 674 nm (Hudson et al. 2008). This yielded a correction factor of 3.49. It should be kept in mind that taking the mean mass-weighted diameter from another study leads to a high uncertainty, given that the size distribution might well be different from M11.

- For the newly added quartz curves we could not infer the total surface area to total mass ratio. Given that quartz is somewhat similar to feldspar however, we use the same factor of 3.5 also for quartz. This comes with high uncertainty as well, as the size distribution and also the particle shape of quartz are likely to be different from K-feldspar. The area shown in the plot covers the $n_{s,geo}$ curves calculated from the $n_{s,BET}$ curves from Zolles et al. 2015 and that calculated from the $n_{s,BET}$ curve for quartz from A13 and like the other areas stretches from the original $n_{s,BET}$ curves to the calculated $n_{s,geo}$ curves.

We have added the following paragraph on p.11, l.7-22 of the revised manuscript:

"*For comparison, the $n_s$ fits for K-feldspar from Atkinson et al. (2013), for kaolinite KGb-1b from Murray et al. (2011), for Illite NX from Broadley et al. (2012), data for quartz from Atkinson et al. (2013) and Zolles et al. (2015) and the $n_s$ parameterization curve from Niemand et al. (2012) are shown. The K-feldspar, kaolinite, illite and quartz curves and data points were provided as $n_{s,BET}$, i.e. the surface area of the particles was measured with the Brunauer, Emmett and Teller (BET) nitrogen adsorption method (Brunauer et al., 1938). This method yields typically a higher surface area than that based on volume equivalent diameter. The literature $n_{s,BET}$ was converted to $n_s$ using a conversion factor of 3.5 in case of K-feldspar as given in the supplementary material of Atkinson et al. (2013). For illite, we followed Hiranuma et al. (2015), using a specific surface area (SSA) of 104.2 m² g⁻¹ (Broadley et al., 2012) and a ratio of total surface area to total mass of 6.54 m² g⁻¹ (Hiranuma et al., 2015). Similarly, for kaolinite we used SSA = 11.8 m² g⁻¹, a density of 2.63 g cm⁻³ and a mean mass-weighted diameter of 674 nm (Hudson et al., 2008), yielding a correction factor of 3.49. For the same $n_{s,BET}$ values, the three quartz samples from Zolles et al. (2015) were active over a range of 10 K. No SSA values were provided therefore we used also a conversion factor of 3.5 given that feldspar is somewhat similar to quartz. This comes with a very high uncertainty, as the size distribution and particle shape of quartz are likely to differ from the K-feldspar of Atkinson et al. (2013). The K-feldspar, kaolinite, illite and quartz $n_s$ areas cover the range from the $n_{s,BET}$ as provided in the literature and the calculated ns to show the uncertainty inherent to the conversion. It can be seen that all desert dust samples fall between the K-feldspar and the clay mineral and quartz fits at all temperatures.*"

*3. Your ns vs. Niemand's ns: The parameterization described in Niemand et al. (2012; N12 hereafter) is essentially based on the observed ice number normalized to the measured "total" surface area (see Eqn. 1-4 in N12), while your parameterization introduces another approximated parameter, the weighted mean aerosol surface area (Ave,w; P6 L11). Are these two ns values apples-to-apples? The performance comparison of IMCA-ZINC-IODE to the AIDA chamber, where the N12 study was conducted, with a*

*reference IN material seems available in H15. The authors may extend the discussion given in H15 a bit more to justify the use of Ave,w in its first appearance.*

The weighted mean aerosol surface area is the area corresponding to the mean of the surface area distribution. We chose to use this value because it is impossible in our case to use the total surface area, due to the fact that the frozen fraction *FF* is always larger than 0.1. Therefore Equation 2 of Niemand et al. 2012 cannot be approximated and we would have to consider each size bin separately. This on the other hand is not possible because we don't know which particles acted as INPs. Therefore we use the mean of the surface area distribution (which we call weighted mean aerosol surface area), which is the most comparable value to the Niemand et al. 2012 parameterization.

We agree with the reviewer that the wording "weighted mean aerosol surface area" which we used to distinguish the area from the area calculated by the mean diameter of the number concentration distribution, was confusing. We are now simply referring to it as "mean particle surface area".

Furthermore, we have rephrased p.6,l.9-11

"The surface area-weighted mean diameter ($d_{ve,w}$) was calculated from the resulting fit for each sample(see Table 1) as well as the surface area corresponding to this weighted average diameter ($A_{ve,w}$)."

to now p.6,l.22-23 of the revised manuscript:

*"The mean particle surface area ($A_{ve,w}$) was calculated from the resulting fit for each sample (see Table 1) as well as the corresponding surface area-weighted mean diameter ($d_{ve,w}$)."*

*In addition, the general discussion of the ns concept seems scattered over the multiple sections and somehow cumbersome (i.e., P11 L3-6, P9 L 4-7 and P10 L12-16). For clarity, I suggest the authors to briefly describe the following two things within one section (any):*

*1) Three assumptions of the ns parameterization, which is relevant to the immersion freezing characterization, include i) the probability of ice nucleation is proportional to the available surface area of immersed aerosols, ii) ice nucleation active sites are uniformly distributed over individual particle surfaces and iii) ice nucleation occurs at specific site in a deterministic manner (predominantly T dependent).*

We have added on p.9,l.9-12 of the revised manuscript:

*"Earlier studies have shown that the probability of a particle to act as INP scales with the surface area of the particle immersed in a droplet (Archuleta et al., 2005; Welti et al., 2009; Kanji and Abbatt, 2010). So-called ice-active sites (Vali, 1966) are assumed on the surface of an INP in the deterministic concept (Langham and Mason, 1958). The probability of such a site to be present on a particle increases with the surface area."*

And rephrased p.9,l.4-5

"It should be kept in mind that the assumption that ns stays constant with particle size most likely has limitations for complex polymineral samples such as desert dust particles."

to now p.9,l.21-22 of the revised manuscript:

*"The assumption that active sites are uniformly distributed over individual particle surfaces, and therefore that $n_s$ stays constant with particle size, most likely has limitations for complex polymineral samples such as desert dust particles. "*

*2) The use of the weighted mean aerosol surface area (Ave,w) is something unique in this study (and different from N12).*

As described above, we have rephrased p.6,l.9-11 to now p.6,l.22-23 of the revised manuscript to avoid any confusion:

"The surface area-weighted mean diameter ($d_{ve,w}$) was calculated from the resulting fit for each sample(see Table 1) as well as the surface area corresponding to this weighted average diameter ($A_{ve,w}$)."

*"The mean particle surface area ($A_{ve,w}$) was calculated from the resulting fit for each sample (see Table 1) as well as the corresponding surface area-weighted mean diameter ($d_{ve,w}$)."*

*Accordingly, I suggest revising the conclusion (P 16 L15-17).*

As described above, the way we calculated $n_s$ is the method most comparable to N12. Therefore, we leave the conclusions unchanged.

*4. The effect of milling: The discussion regarding the effect of milling is misleading. Specifically, the reviewer's concerns are as follows:*

*P1 L18-20: This statement is misleading as it sounds like the milling process generally deteriorates IN efficiency of any composite materials. In fact, this statement contradicts to the IN results of the Atacama samples (ns,milled > ns,sieved) presented in Sect. 3.4. (i.e., P14 L29-30). I suggest rephrasing this sentence to be more specific (adding the "may" word would not help).*

We have rephrased the sentence, to now (p.1, l.18-21 new manuscript):

*"Furthermore, we find that under certain conditions milling can lead to a decrease in the ice nucleation ability of polymineral samples, due to the different hardness and cleavage of individual mineral phases causing an increase of minerals with low ice nucleation ability in the atmospherically relevant size fraction."*

*P15 L11-13: Without any evidence of the alternation in defect densities on the Isralel dust surface or calcite surface per milling, this statement (morphology vs. mineralogy) seems too ambitious/strong. It is likely the micro pores on the surface (Subramanyam et al., 2016, Appl. Mater. Interfaces, doi:10.1021/acsami.6b01133) that enhance the IN activity of particles due to the inverse Kelvin effect (Marcolli et al., 2014, ACP). If calcite etc. is breaking up but maintaining smooth surface, there is no reason for its fragments to enhance IN activity. The authors may be aware, but the nature of active sites*

*is still uncertain and under investigation. Concerning these points, I suggest the authors to soften the tone of this statement.*

We believe that we have made clear that this statement is valid only for the Israel sample and we do not generalize this finding (p.15,l11-12 of the original manuscript: "Thus, we conclude for the Israel sample that a morphology effect is small in comparison to the change in mineralogical composition caused by the milling in the analyzed size range."). Furthermore, we have stated that we were not able to investigate properties like the defect density, surface irregularities etc. and base our conclusion on the finding of earlier studies which all found an increase in ice nucleation activity with an increase in defect density etc. Since the Israel sample lost ice nucleation activity with milling, an increase in defect density therefore is unlikely (except if defects are not really active sites). At least its effect on ice nucleation activity must be small. Hence, the change in mineralogy, which is expected for the sub 2.5 µm particles, likely explains the increase in ice nucleation activity. We have modified the respective sentence to make it more clear that we only refer to the Israel sample in this case. p.16,l.4-6 of the revised manuscript:

"*As milling reduced the ice nucleation activity for the Israel sample, we conclude for this specific sample that any morphology effect is small in comparison to the change in mineralogical composition of the analyzed size range caused by the milling.*"

*5. Airborne vs. Surface: To me, the essence of this paper is summarized in P10 L9-10. This single statement and the method to reach this point would be worth a paper. The authors have already provided a great conclusion (P16 L8-10 & P1 L9-10). Focus your story along with this line. I have some questions regarding airborne dust vs. surface-collected dust as follows:*

We thank the reviewer for the remarks above.

*P11 L9-10: Interesting. The authors are right - according to Table 4, airborne samples in general seem having larger a (0.48 ± 0.07 K-1) as compared to the average of the rest (0.38 ± 0.16 K-1), suggesting high T dependency of the airborne samples. But, this is not all about T dependency. This may be rather an indication that the airborne particles miss certain active sites that can be activated at high T (owing to the difference in mineralogy???). The authors may discuss and clarify this point. This observation seems important.*

We agree with the reviewer that a steep $n_s$ curve, having a high $a$ value in Table 4 can, in the active site concept, be described as missing certain active sites which activate at warm temperature. These two aspects (warm $T$ active sites and $T$-dependency) do not contradict each other but are descriptions of the same observations. We have added on p.11,l.32-33 of the revised manuscript:

"*This can been seen as an indication of active sites, which activate at warmer temperatures, being more frequent on the surface-collected samples compared to the airborne samples.*"

*P12 L26-27: This is another important statement. Elaborate a bit further by discussing the atmospheric relevancy of the minerals uniquely found in the airborne samples. Put clear emphasis if your bulk measurements at least suggest the atmospheric relevance of quartz/K-feldspar. According to P7 L14-15*

*and P7 L20-22, I have a feeling that there would be reduced amount of quartz/K-feldspar in air. Any comments? This argument seems the core research question of this manuscript.*

All airborne samples stem from North Africa whereas the surface-collected samples are from sources all over the globe. Hence, the direct comparison of airborne to surface-collected samples might be biased by location dependent differences in mineralogy. This is why we don't focus on this comparison alone.

We have discussed in the original manuscript that harder minerals such as feldspar and quartz are more likely to be found in larger grain sizes whereas softer minerals such as calcite and clays tend to be found more often in the small grain sizes. (P.3, L.30-31). Furthermore, we have described in the introduction the mineralogical composition of airborne dust samples over the North Pacific, Morocco and Israel, all indicating that quartz and feldspar are commonly found in atmospheric dust particles (p.3,l.34-p.4,l.6).

To stress the atmospheric relevance of quartz further, we have added on p.3,l.26-27 of the revised manuscript:
"*Quartz is commonly (5-50 wt%) found in atmospherically transported Saharan dust samples (Avila et al., 1997; Caquineau et al., 1998; Alastuey et al., 2005; Kandler et al., 2009).*"

Our analysis of the six samples where the mineralogy analysis was directly representative of the particles on which we measured ice nucleation, i.e. which were in the size range smaller than 2.5 μm and therefore also likely to be lifted by wind, suggested also that feldspars plus quartz are the best predictors of ice nucleation activity. Out of these six samples, three were airborne. If the analysis is done on the airborne samples alone the same trend is seen.

We further stress the importance of feldspar and quartz even in low concentrations on p. 15, l.9-10:

"*However, our results indicate that feldspar or quartz present in the bulk dust will dominate its freezing behavior down to 238 K.*"

and in the conclusions on p.16, l.21-28:

"*Keeping in mind that quartz is ubiquitous in atmospheric desert dust, this suggests that quartz plays a more prominent role for atmospheric ice nucleation than previously thought. The clay mineral (illite and kaolinite) and calcite content of the dust samples negatively correlated with ns at all studied temperatures, suggesting a minor importance of these minerals for the ice nucleation activity of natural dust samples in the immersion mode, especially if quartz or feldspar are present. Atkinson et al. (2013) suggested that the global mineral dust INP concentration down to a temperature of about 240 K is dominated by feldspar. At temperatures between the homogeneous freezing limit and 240 K, where quartz is an active INP, it dominates the total INP concentration as it is much more abundant than feldspar. Our experiments on natural dust confirm this suggestion.*"

and on p.17, 1-4:

"*For all desert dust samples we found a high correlation of the ice nucleation activity of particles smaller than 2.5 μm with the quartz content of the dust samples. This shows that despite the dominance of the*

*clay minerals in the small size fraction, quartz is an important atmospheric INP component and also found in the particle size fraction with the longest atmospheric residence time.* "

*Table 3: It is really bothering me that the Tenerife sample (6% microcline!) is not showing any superb IN behavior as compared to other airborne dusts. The authors said that the mineralogy inferred by XRD is representative of the aerosolized Tenerife sample (P7 L28-30). The authors disregard the contribution of atmospheric processes (P14 L17-19). Then, what is limiting the IN of this particular dust?*

Mineralogy analysis is a bulk measurement, hence we cannot infer the mixing state and therefore don't know, if a) 6% of all particles are pure microcline particles or b) all particles contain 6% microcline or c) an intermediate mixing state. In case a): even if all microcline particles would activate as INP, this would not reach the lower detection limit of IMCA (10 %). In case b): it may well be that the 6% microcline aren't on the surface of each single dust particle but covered by less active minerals (e.g. Zhu et al 2006). For case c) both effects could play a role.

6 % microcline is clearly at the lower end of K-feldspar content in our samples. An effect on ice nucleation from the 6 % of microcline should be detectable at warmer temperatures but unfortunately no FRIDGE measurements are available for this sample. The microcline fraction is included in the K-feldspar fraction used for the correlations in section 3.3.

*6. Table 5 (P13 L5-P14 L24): This part includes a number of flaws (i.e., ns, geo vs. ns, BET) and needs substantial improvements. In fact, with given limitations/assumptions in P14 L6-24, I am not convinced that these 'relative' correlations add much meaning to the manuscript. Do the authors really require this part to draw their conclusion? This whole statistics part of the manuscript could be deleted?*

The conversion of the literature $n_{s,\,BET}$ to $n_{s,\,geo}$ has no effect on the correlations because the whole analysis was consistently done using $n_{s,\,geo}$. We believe that this approach of directly correlating the mineralogical components with the ice nucleation ability ($n_s$) is a novel approach and the results are valuable. We have added the correlation plots in the supplementary material and have added if the correlations are statistically significant in Table 5 and the new Table 6. We agree that the method has limitations due to the low number of samples and the comparison of a volume property (mineralogy) to a surface property (ice nucleation ability). Therefore, the correlations are often statistically not significant. Both limitations have already been discussed in the original version of the manuscript. Despite these limitations, the results shed new light on the importance of mineralogy and in particular of quartz for ice nucleation of atmospheric dust. We believe this part is clearly required to draw the conclusion that quartz seems to play a more important role for ice nucleation of natural desert dust samples at the investigated temperatures than believed so far and much more than do clays or any other minerals apart from feldspar. This has been suggested by Atkinson et al. 2013 based on mineralogical maps and $n_s$ of single minerals. Our study confirms this now for airborne and surface-collected natural dust samples. Since single mineral studies of the ice nucleation ability of quartz have yielded inconsistent results and focus has been placed on the role of clays and in recent years on feldspar, this is an important new finding. We have added on p.16, l.25-28 of the revised manuscript:
"*Atkinson et al. (2013) suggested that the global mineral dust INP concentration down to a temperature*

*of about 240 K is dominated by feldspar. At temperatures between the homogeneous freezing limit and 240 K, where quartz is an active INP, it dominates the total INP concentration as it is much more abundant than feldspar. Our experiments on natural dust confirm this suggestion.* "

*Again, I generally agree that quartz and microcline are IN active and may have potential importance in the atmospheric IN. What would be more valuable to see is if there is any 'absolute' relation between mineralogy and IN. For example, the authors may explore if the natural dust ns (or FF) can be optimized/predicted by its composition and associated individual ns scaled to the surface fraction (SF) of each component (x1,x2 ... to xi) [i.e., ns,dust = (ns,x1 x SFx1) + (ns,x2 x SFx2) + ... + (ns,xi x SFxi)]. Note that H15 attempted, but no success. Give more in-depth thoughts regarding the role of particular mineral IN.*

This is an interesting suggestion. Please note that the method of correlating the sum of minerals that are assumed to contribute to the ice-nucleation activity at the investigated temperatures with $n_s$ is a simplified version of the proposed method. However, we do not have surface specific mineralogical information, therefore no surface fraction of each component can be determined. As a first attempt one could assume surface fraction to equal volume fraction (see O'Sullivan et al. 2014). But given the uncertainty related to the surface-specific mineralogy, the conversion from $n_{s,\,BET}$ to $n_{s,\,geo}$, the polydisperse size distribution of our sample, the wide spread of $n_s$ of minerals that are supposed to be the same within one single study (e.g. plagioclase and K-feldspar in Harrison et al. 2016, quartz in Zolles et al. 2015) or between studies (comparison provided in Harrison et al. 2016 of their own study and Zolles et al. 2015, Emersic et al. 2015 and Atkinson et al. 2013 for feldspars) and the failed attempt of H15 for a rather simple dust sample, we believe that the method proposed by the reviewer is too ambitious to give meaningful results for our dust samples at the moment. This however would be an excellent suggestion for the future, when monodisperse natural dusts are measured with the same instrument/method at the same temperatures as reference minerals and the surface fraction of each component can be determined with reasonable accuracy.

*Specific comments/suggestions*

*Introduction*

*P2 L22: Briefly explain what the authors mean for 'contradicting results'.*

We have replaced

"The IN ability of soot (Brooks et al., 2014; Kulkarni et al., 2016) and secondary organic aerosol (Prenni et al., 2009; Ignatius et al., 2015) at heterogeneous freezing temperatures is still debated as contradicting results have been observed."

with (p.2,L.21-25 of revised manuscript):

*"The ice nucleation ability of soot (Brooks et al., 2014; Kulkarni et al., 2016) at heterogeneous freezing temperatures is still debated as contradicting results were observed, spanning from hardly any ice nucleation ability at T > 236 K (Kanji et al., 2011) to up to 3 % of soot particles active in the immersion mode (DeMott, 1990). Similarly, the reported freezing behavior of secondary organic aerosol particles varies from inefficient to comparably efficient (Möhler et al., 2008a; Prenni et al., 2009; Wang et al., 2012; Ladino et al., 2014; Ignatius et al., 2016)."*

*P2 L24: Briefly describe 'certain minerals'.*

"Recently, also certain minerals have been identified to nucleate ice at temperatures up to 271 K (Harrison et al., 2016)."

has been replaced by (p.2,l.27-28 revised manuscript):

"*Recently, the K-feldspar microcline and the Na-feldspar albite, both minerals found in atmospheric dust, have been identified to nucleate ice at temperatures up to 271 K (Harrison et al., 2016).*"

*P2 L25: "For the implementation…" - I suggest starting a new paragraph here regarding the IN parameterizations. This way, the previous paragraph (L18-25) reads more like a general introduction of atmospheric INPs and their diversity (biological and non-biological).*

done

*P3 L19: The authors may mention that the abundance of quartz in atmospheric dusts is consistently high (i.e., ~10% in volume) in the size range of ~1 to 35 µm geometric diameter (see Table 1 of Kandler et al., 2009, Tellus; cited in this paper), which would add the atmospheric relevance and general importance of quartz. Such information could also fit in P4 L1-3.*

We added (p.3,l.26-27 of the revised manuscript):

"*Quartz is commonly (5-50 wt%) found in atmospherically transported Saharan dust samples (Avila et al., 1997; Caquineau et al., 1998; Alastuey et al., 2005; Kandler et al., 2009).*"

and (p.3,l.32-33):

"*Their samples contained quartz fractions between 1 and 26 wt%, K-feldspar fractions between 0 and 10 wt%, and plagioclase fractions between 0 and 22 wt%.*"

*Methods*

*P5 L26-27: Does the size distribution of particles in the chamber change over 3.5 hours (i.e., the filter sampling period; P8 L15-16)? Large particles settle down faster than the smaller ones, and the authors infer that certain mineral compositions are large in their sizes (e.g., P15 L33). Please clarify and discuss potential consequences, if any.*

We state on p.6, l. 11-13 of the original manuscript (now p.6, l.14-16), and in Table 1, the mean surface area varied by 6-24 % over the course of a day of experiments (which lasted typically between 5-12h), except for the Great Basin sample (64 %). Since the 3.5h filter sampling period is lower, the variation is also smaller. Nevertheless, this maximum error has been used when calculating the uncertainty of $n_s$ (see p.10, l.18-19 of original and p.10, l.21-22 or revised manuscript). Since the uncertainty resulting from this decrease in surface area is small compared to the uncertainty in the INP concentration, we do not discuss potential consequences. We have added on p.10,l.29-30 of the revised manuscript:
"*The error bars in $n_s$ are derived by error propagation from the error in FF and ($A_{ve,w}$) and are dominated by the error in FF.*"

*P5 L28: This background (~10%) seems high for the IN research. Any justification or measurements that this much background particles have no substantial contributions to heterogeneous freezing?*

We agree that the value is comparably high. But since we only consider frozen fractions between 10-90% in the IMCA-ZINC experiments, we have no influence of the background in our analyses even if all background particles would act as INP and none of the sample particles were INPs. Furthermore, it is expected that the residual particles are of the smallest sizes and hence least ice nucleation active. Nevertheless, in case of the FRIDGE measurements there could be an influence even though the observed results suggest there was hardly any: The order of measurements was Egypt, then Atacama milled, then Etosha, and at last Taklamakan. The Atacama milled sample had an order of magnitude higher $n_s$ than the other samples. So, the influence of residuals from the Egypt sample on the Atacama milled measurements (which would decrease the Atacama milled $n_s$) would be less than 1 % if the $n_s$ of the Egypt sample residuals was the same as during the actual experiment. Due to their higher $n_s$, residual particles from the Atacama milled sample could have increased the observed $n_s$ of the Etosha sample, even though less than 6 % of the particles were residuals. This could potentially explain why we found a comparably high ice nucleation activity which cannot be related to any of the known ice nucleation activities of the mineral components. As shown in Figure 2 of this document however, the ice nucleation activity at 240-242 K seems to be related to organic/biological material. This does not rule out a large influence of the Atacama milled residual particles on the Etosha sample in the FRIDGE measurements but makes it unlikely.

The Etosha and the Taklamakan sample finally show similar $n_s$ in the FRIDGE data. This is in line with the results from the IMCA-ZINC measurements at lower temperature, suggesting that there is only negligible influence if any.

*P5 L30: Briefly describe "coincidence effects" or cite proper papers.*

We have added (p.6, l.12 of the revised manuscript)

"*For the IMCA-ZINC measurements the particle concentration was diluted to about 60 cm$^{-3}$ to avoid coincidence effects in the detector which occur if more than one particle is present in the laser beam of the detector Nicolet et al. 2010.*"

*P7 L2: According to Table 3 of K16 (Kaufmann et al.), the Israel sample contains some sanidine. Moreover, sanidine in the natural surface samples seems non-negligible as 6 out of 12 natural surface samples examined in K16 shows the presence of sanidine. Was your sample totally different from K16? Or the the sample may not be completely homogeneous in terms of mineralogical distribution even within a same batch? Was the XRD data interpretation method somewhat different from K16? Please clarify.*

The Israel and the Namib (Etosha) samples are the same as studied by K16 as we stated on p.5, l.7-8 (now p.5, l. 17 of the revised manuscript): "*The Israel sample and the Namib sample are from the same batch as those studied in Kaufmann et al. (2016).*"

The XRD data interpretation method was done slightly differently from K16. K16 used in addition to AutoQuan the software EVA, whereas we used AutoQuan and only for unidentified minerals used EVA. The Israel sample only contains about 2 % K-feldspar. Due to this low fraction and since the feldspar

analysis is one of the most challenging part of the mineralogical composition analysis of a polymineral sample, the different results are reasonable.

We have replaced (p.7,l.2 of the old manuscript):

"No sanidine feldspar was found in any of the samples."

with (p.7,l.15-17) of the new manuscript:

"*In case of a low K-feldspar content of a few wt% it was not possible to determine if K-feldspar was present as microcline, orthoclase or sanidine or a mixture of the different phases. Values are given for the K-feldspar with the best Rietveld fit result.*"

*P7 L20-22: Doesn't this just mean that a majority of large particles (up to 32 µm) break up by the RBG milling? I mean that RBG may do more than just aerosol dispersion, correct? Long story short, is it really fair to assume that those aerosolized particles are identical to the sieved bulk used for XRD as the authors mention in P13 L20-23? Further clarification seems necessary. Accordingly, the authors may consider rephrasing the relevant text in the conclusion (i.e., P15 L29-32). 2.5 µm sounds like a magic number as it is right now.*

The RBG may deagglomerates but does not break up or grind particles as the residence time is low and it is a stainless steel brush not a mill or balls. Even if particles would break up during generation, their mineralogical composition would remain the same. If the mineralogical composition is representative depends only on the fraction of particles that make it into the tank reservoir compared to those that are left behind in the cyclone. This fraction has been semi-quantified as described in the manuscript.

The 2.5 µm is simply the $D_{50}$ of the cyclone used during aerosol generation (p.5, l26 of the original manuscript). There is nothing magical about this number, but it is the size cut-off we used consistently for particle generation.

*P7 L28-30: For the reason given above, I am not sure if the authors can asset like this.*

We hope we could convince the reviewer of the plausibility of the method.

*Results & Discussion*

*P10 L3: The homogeneous freezing regime presented in this manuscript is based on CNT? Or anything different? I suggest adding proper reference(s) here at least.*

The homogeneous freezing regime indicated in Figure 4. a) - c) is derived from CNT for 10 µm droplets using formulae and constants given in Ickes et al. 2015. Reference experiments on homogeneous freezing with the IMCA-ZINC setup can be found in Hoyle et al. 2011 Fig.5. We add a reference to Hoyle et al. 2011 and Ickes et al. 2015 to the figure caption.

*P10 L9-10: I encourage the authors to clearly state that the same trend holds true for another metric, ns(T).*

We have added on p.10,l.33-34 of the revised manuscript:

"*Like their FF, the range of $n_s$ of the Saharan samples is comparable to those of the non-Saharan ones (Fig. 4f).*"

*P11 L2-3: It looks to me that the Niemand parameterization falls in the middle of your 15 ns(T) spectra. Please clarify what "rather at the lower end" means quantitatively.*

We have added on p.11,l.26 of the revised manuscript as subclause:

"*(a factor of 3 to 4 below the average $n_s$(T) of all measured curves, not shown).*".

*P11 L13-14: The word "overpredicts" implies that the Niemand parameterization (N12) is wrong. The authors may want to soften the tone and rephrase. Your assumption vs. assumption made in N12 would be discussed here.*

We have rephrased the sentence to (p.12,l.1-3 of the revised manuscript):

"*The parameterization from Niemand et al. (2012) predicts one to two orders of magnitude higher $n_s$ than measured by FRIDGE. Only for the Atacama milled sample at T = 251 - 256 K the parameterization shows about 30 % higher values than the measurements.*"

We believe that a parameterization cannot be right or wrong. At first it is simply representative for the samples which were used to define it. Even for those, there is an uncertainty related to the parameterization because of the scattering of the data points included. It must be tested if the parameterization also represents other samples and if not, why not. This is what is done in our manuscript.

*P11 L11-20: I do not find the scientific significance of having FRIDGE data included in this manuscript. The authors briefly discuss about the FRDIGE results in this part and only a bit more afterward. The reviewer does not find that the FRIDGE results are complementing IMCA-ZINC-IODE, vice versa. In addition, no proper justification for why FRIDGE was conducted for a subset of samples is provided. Does the authors' conclusion change without the FRIDGE data?*

The scientific significance of having FRIDGE data included is given on p.4, l.18-19 of the original manuscript:

"This allowed examination of immersion freezing at temperatures between 250 and 262 K, hence covering a wide range of heterogeneous freezing temperatures."

We have rephrased it to p.4, l.31-32 of the revised manuscript:

"*This allowed examination of immersion freezing at temperatures between 250 and 262 K, covering a wider range of heterogeneous freezing temperatures than would otherwise be possible with IMCA-ZINC alone.*"

IMCA-ZINC is not sensitive enough to measure *FF* below 10 % reliably. Therefore, FRIDGE was used to extend the covered temperature range. This was not initially planned as a part of the dust measuring

campaign and for logistical reasons, the particle collection for FRIDGE was only done at the end of the campaign. Therefore only four samples were collected. We don't see a value of adding this information to the manuscript.

The authors' conclusions on K-feldspar determining mainly the ice-nucleation behavior of natural dust at T>250K (p.15, l.18-20 of the original manuscript) is entirely based on the FRIDGE data. We therefore see a high value in keeping the FRIDGE data.

*Fig. 5: Adding quartz reference spectrum (Atkinson et al., 2013) would be nice.*

We did not show a reference spectrum for quartz in the original manuscript because different authors found very different results even within a single study (Zolles et al. 2015, Atkinson et al. 2013). However, since we are now showing areas of $n_s$ than lines, we have included an area of $n_s$ for quartz.

*Fig. 5 Cont'd: Having another panel depicting the highest-median-lowest spectra of airborne ns vs. those of surface-collected ns would add some clarity and strengthen the paper.*

We agree, that's a great suggestion. We have added such a Figure (6a) in the revised manuscript. Furthermore, we have added the following paragraph on p.12,l.9-18:

"*Figure 6a shows the median and minimum to maximum $n_s(T)$ range of the airborne and surface-collected samples. This illustrates that the $n_s$ range of the airborne samples falls in the lower half of the $n_s$ range of the surface-collected samples or even below. It shows that for immersion mode ice nucleation surface-collected dust samples are not representative for airborne dust samples, which all stem from North Africa, the world's largest source of atmospheric dust. This might be caused by a non-representative surface-dust collection, e.g. soil rather than dust is collected which has a different size distribution and composition, or dust from a location where threshold wind velocities for dust lifting are not reached. Another cause could be that atmospheric processes taking place during or after particle lofting may alter the particle surface and decrease the ice nucleation ability which has been suggested to occur in the field (Cziczo et al. 2013) and laboratory (Sullivan et al. 2010, Augustin-Bauditz et al.2014). The potential effects of mineralogy on the ice nucleation activity at different temperatures is investigated in the following section.*"

*P12 L4-9: The discussion given here makes the review think something other than minerals (e.g., P5 L15-17) are competing for IN at given T range. The reviewer is aware that the focus of the current work is on mineralogy vs. IN (and no biological INP perspective at all). That said, the authors should extend the discussion regarding the potential bioaccessibility of dust surface (Augustin-Bauditz et al., 2016, ACP; O'Sullivan et al., 2016, ACP) and other IN species that might be present in soil (O'Sullivan et al., 2014, ACP; Tobo et al., 2014, ACP; Pummer et al., 2015, ACP). Such information would strengthen the manuscript.*

We agree with the reviewer's comment. We performed parallel measurements of particle fluorescence as a proxy of biological material, but due to technical reasons these data were not useable. We were able to repeat some of the deposition/condensation measurements, which will be included in the part 2 paper associated with this manuscript (Boose et al., 2016). The experiments were performed using the portable ice nucleation chamber PINC with the original samples as well as samples heated to 300°C (573 K) for 10h. PINC, which works like ZINC but is shorter, has a shorter residence time and sampled particles which were dry rather than pre-activated. Therefore, the results are not necessarily comparable to the immersion mode results from the current manuscript because full droplet activation prior to freezing is not guaranteed.

Figure 2 shows the condensation mode $n_s$ at 240 and 242 K and at the highest $RH_w$ sampled (100 – 105 %) to mimic immersion freezing (Hiranuma et al. 2015). Filled symbols refer to unheated, open symbols to heated samples. Heating had little to no effect on the ice nucleation activity of the Australia, the Atacama milled and Peloponnese samples. The Etosha sample lost most of its ice nucleation activity after heating, which is the sample for which we could not relate the comparably high ice nucleation activity to its mineralogy as it consisted mostly of calcite and dolomite, which have a low ice nucleation activity (Atkinson et al. 2013, Kaufmann et al. 2016), and ankerite, of which the ice nucleation activity is not known (Kaufmann et al. 2016). This is consistent with the manuscript conclusions that the ice nucleation activity is mainly caused by the mineralogy of the dust samples. The situation may be different at warmer temperatures but unfortunately we cannot repeat investigations at these conditions due to the small remaining sample size. The full data from these additional measurements and further analysis will be shown in the second paper of the series (Boose et al. 2016).

[Figure]

Figure 2: Condensation mode $n_s$ for selected heat treated samples.

In the revised manuscript we have added a paragraph on non-mineral matter mixed with the dust and its potential effects on the ice nucleation ability of the dust (p.4,l.15-20 of the revised manuscript):

*"Non-mineral matter, which can become internally or externally mixed with the mineral dust before or after emission, may affect the ice nucleating behavior of the dust. Sulfuric acid (Sullivan et al., 2010; Augustin-Bauditz et al., 2014) or secondary organic aerosol coating (Möhler et al., 2008a) has been observed to decrease the ice nucleating ability while exposure to ozone (Kanji et al., 2013) or the presence of ammonium sulfate (Boose et al., 2016b) has been suggested to improve it. Biological material can adsorb to mineral dust, enhancing its ice-nucleating ability (Schnell, 1977; Conen et al., 2011; O'Sullivan et al., 2016)."*

We have rephrased p.5,.15-16 (now p.5,l.28-30 of the revised manuscript):

"All natural dust samples are expected to be very heterogeneous, i.e. external and internal mixtures of different minerals and potentially biological material (Meola et al., 2015)."

*"The composition of natural dust samples is presumed to be heterogeneous, i.e. external and internal mixtures of different minerals and potentially containing organic or biological material (Meola et al., 2015). "*

We have replaced p.12,l.8-9 (now p.13,l.7-9 of the revised manuscript):

"No study so far has investigated the IN behavior of pure ankerite. Thus it remains unclear what leads to the observed high IN acitivity at T < 242 K of the Namib sample."

*"Thus, the high ice nucleation activity at T < 242 K of the Etosha sample is not explainable by the known ice nucleation ability of its mineral components. To our knowledge, no study so far has investigated the ice nucleation behavior of pure ankerite."*

We have replaced p.14, l.22-23:

"If this is due to reasons other than mineralogy such as coating as suggested by Kaufmann et al. (2016) or if the present minerals ankerite, dolomite or muscovite can lead to a high IN activity at T < 243 K under certain circumstances is not known."

with (now p.15,l.15-16 of the revised manuscript)

*"Measurements in the condensation mode, which are the subject of part 2 of this study, suggest that the ice nucleation activity of this sample is in large part related to organic or biological material mixed with the dust."*

*Conclusion*

*P15 L18-19: Which data infers the FF and $n_s(T)$ results at temperatures above 250 K? I do not see them in Fig. 4. FRIDGE?*

The $n_s$ of FRIDGE at T>250K is shown in Fig 5b). We have deleted "the FF" on p.15, l.19 (original manuscript). The sentence reads now (p.16,l.12-13 in the revised manuscript):

*"The comparison showed that at temperatures above 250 K the highest $n_s$ is related to the highest fraction of K-feldspars in the sample..."*

*P16 L2-4: Your data presented in Tables 2 and 3 (i.e., Atacama milled vs. sieved; Israle milled vs. sieved) seem contradicting to this statement. For instance, I do not see any increase in the quartz fraction.*

It is correct that for the Israel and the Atacama sample, the milling did not lead to an increase in the quartz fraction. This has the following reasons: For the Israel sample, the milled and the sieved sample should have the same mineralogy because the batch which was already sieved to below 32 μm was further milled. In the case of the Atacama sample, were the original pre-processed sample was milled and compared to the < 32 μm sieved fraction, the quartz fraction actually is slightly lower in the milled compared to the sieved sample. However, one can see that the K-feldspar (orthoclase) fraction is 10.5 % higher than in the sieved sample. As feldspar is also hard and in addition due to its reactivity less common in the small particle fraction, the milling increased the feldspar fraction, resulting in a reduction of the quartz fraction.

We refer here mostly to the Morocco and Australia samples which had almost no particles in the size range < 32 μm and therefore needed to be milled. They consisted both mainly of quartz and feldspar.

*P16 L6-8: This part seems contradicting to the previous statement (that is, P15 L33).*

We have now changed the sentence on p.15,l.33:

"Quartz is a comparably hard mineral and thus less common in the smallest dust size fractions. Since we measured IN activity of particles smaller than 2.5 μm and found a high correlation with the quartz content of the dust samples we show that quartz is nevertheless an important atmospheric INP component "

to p.17,l.1-4 of the revised manuscript:

*"For all desert dust samples we found a high correlation of the ice nucleation activity of particles smaller than 2.5 μm with the quartz content of the dust samples. This shows that despite the dominance of the clay minerals in the small size fraction, quartz is an important atmospheric INP component and also found in the particle size fraction with the longest atmospheric residence time."*

With this change we now in the revised manuscript want to say that one could expect that quartz should be only found in the larger size fraction of dust particles because of its hardness (rather than stating that as a fact as previously done so). We then follow this with a discussion demonstrating that in our results quartz can be indeed found in the atmospheric size fraction and can be important to predicting ice nucleation behavior.

*Technical comments/suggestions*

*P1 L1-2: "Traditionally, clay minerals were assumed to determine…"* ➔ *"Since natural dusts are composite in nature, clay minerals were typically used as a proxy to determine…"*

We prefer to leave the sentence as it is. We base this statement on early publications on ice nucleation, e.g. Kumai 1961. Furthermore clays were used as a proxy because clays can form a substantial component of natural dust.

*P1 L14: "…activity in a given sample above 253 K that can be attributed to…"*

We prefer to keep the wording "at 253 K" because we did our analysis of correlating $n_s$ with mineralogy only at 253 K and not at warmer temperatures. Hence, "above 253 K" would be speculative.

*P1 L17 and hereafter: Use the Italic font for T throughout the manuscript.*

done

*P2 L3: "determine" ➔ "influence" - Besides primary ice nucleation, secondary ice processes can also contribute to the lifetime of clouds.*

done

*P2 L20: Pummer et al. missing as INM references*

"*Pummer et al. 2012*" was added

*P2 L21: DeMott et al. missing as soot IN references*

"*DeMott et al. 1990*" was added.

*P2 L32: ", respectively"*

A comma was inserted.

*P2 L32-P3 L1: Too many things packed in a single sentence. Split into two sentences.*

done (p.3, l.2-4 of the revised manuscript)

*P2 L33: between ➔ amongst*

done

*P3 L13: "K-feldspars (microcline, orthoclase and sanidine) were…"*

done

*P3 L20-22: Redundant (P3 L4-6)*

We have deleted the first half of the sentence (p.3,l.24-26 of the revised manuscript):

"It has been proposed that differences in the surface structure can lead to different IN abilities of quartz samples and it is suspected that functional groups on the surface of feldspars and quartz are responsible for their higher IN ability (Zolles et al., 2015)."

*"It is suspected that functional groups on the surface of feldspars and quartz are responsible for their higher ice nucleation ability (Zolles et al., 2015) but it is unknown where the high variability stems from."*

*P3 L27: Delete "hence". Size and composition are inherently related. Is that what the authors want to say? If so, state so.*

The sentence

"It has been observed that the size distribution and hence mineralogical composition of dust changes during its emission and transport compared to that on the surface (D'Almeida and Schütz, 1983; Murray et al., 2012; Knippertz and Stuut, 2014). The reason for this is a size dependent mineralogical composition caused by differences in the hardness, cleavage and shape of minerals."

has been changed to (p.3, l.34 - p.4, l.2 of the revised manuscript):

*"It has been observed that the size distribution of dust changes during its emission and transport compared to dust on the surface. This leads to variations in the mineralogical composition of the dust, (D'Almeida and Schütz, 1983; Murray et al., 2012; 35 Knippertz and Stuut, 2014) as the mineralogical composition is size dependent due to differences in the hardness, cleavage, shape and reactivity of minerals."*

*P3 L31: No "hence"*

done

*P4 L29: "The GPS coordinates of our collection sites…"*

done (p. 5, 7-8)

*P5 L15-16: "The composition of all natural samples are presumably heterogeneous…"*

done (p.5, l. 28-30)

*P6 L10: "… sample (see …"*

done (p.6, l. 23)

*P6 L15: "… (sieving/milling)"*

done (p.6, l. 29)

*P8 L5: "The cloud droplets"* ➔ *"The simulated cloud droplets" or "The activated droplets"*

We have deleted "cloud" (p.8,l.16 of the revised manuscript).

*P8 L15: "subsequent"* ➔ *"independent"*

done (p.8, l. 29)

*Eqn. 4: Missing negative sign on the RHS of Eqn.4?*

done (p. 9)

The sentence

"It should be kept in mind that the assumption that ns stays constant with particle size most likely has limitations for complex polymineral samples such as desert dust particles."

has been rephrased to (p.9,l.21-22 of the revised manuscript):
"*The assumption that active sites are uniformly distributed over individual particle surfaces, and therefore that $n_s$ stays constant with particle size, most likely has limitations for complex polymineral samples such as desert dust particles.*"

*P9 L14-15: Awkward sentence. Simply say something like our measurements are valid with χ in the range of 1.1 to 1.6.*

We have rephrased this and the preceding sentence

"Since one mode was detected in each instrument's size range, a variation of the shape factor  was tested to check if a better overlap of the two size distributions could be achieved. Within realistic limits ($1.1 \leq \chi \leq 1.6$) for shape factors of atmospheric dusts (Alexander, 2015) the two modes remained distinguishable and are thus assumed to be real."

to (p.9,l.32-p.10,l.3 of the new manuscript):

"*Since one mode was detected in each instrument's size range, the shape factor χ was optimized to give the best overlap of the two size distributions. For any shape factor within realistic limits for atmospheric dusts ($1.1 \leq \chi \leq 1.6$, Alexander 2015) the two modes remained distinguishable.*"

*P9 L21-22: Awkward sentence. Simply say …because of the presence of predominantly large particles… or something similar.*

done (p.10, l.9-10)

*P 10 L12-13: "Due to the heterogeneous and possibly size-dependent particle compositions, a partial step function like activation spectrum could be…"*

done (p.10, l.23-24)

The second "overall" was deleted. (p.11, l. 6)

*P11 L2: "The comparison between the Niemand parameterization and our parameterizations based on…"*

We have rephrased the sentence to (p.11,l.24-26 of the revised manuscript):

*"For T < 250 K the parameterization falls in the lower end of the range of ns observed for our broader collection of global surfacecollected and airborne dust samples (a factor of 3 to 4 below the average ns(T) of all measured curves, not shown)."*

*P11 L5: "polydisperse nature" ➡ "heterogeneous properties"*

We have replaced "nature" with "*size distribution*" (p.11, l.28 of the revised manuscript)

*P11 L23-24: This sentence does not fit in here. I suggest deleting.*

We have added another sentence to create a smoother transition (p.12,l.20-21) of the revised manuscript:

*"By analyzing the bulk mineralogy we investigate if the different natural dust's mineralogical composition explains the observed ice nucleation activity."*

*P14 L34: The authors may want to remind the reader that 2.5 µm is D50 of your cyclone.*

We added "*(the $D_{50}$ cut-off of the particle generation system used)*" after "2.5 µm" on p.16, l.5 of the revised manuscript.

*P15 L10-13: "However… Thus…" - Awkward transition. Rephraze.*

The sentences have been rephrased to (p.16, l.3-6 of the new manuscript):

"However, an increase in defect density and surface irregularities has been shown to increase the IN activity of monomineral or single compound samples. Thus, we conclude for the Israel sample that a morphology effect is small in comparison to the change in mineralogical composition caused by the milling in the analyzed size range."

*"An increase in defect density and surface irregularities has been shown to increase the ice nucleation activity of monomineral or single compound samples. As milling reduced the ice nucleation activity of the Israel sample, we conclude for this specific sample that any morphology effect is small in comparison to the change in mineralogical composition of the analyzed size range caused by the milling."*

*P15 L22 above or below?*

We have deleted the respective sentence.

*P15 L27-28: I do not understand this sentence. Rephrase.*

The sentence

"The size dependent enrichment of different minerals leading to differences in ns highlights the interplay between IN ability and atmospheric relevance of certain mineral phases."

was replaced by (now p.16, l.29):

*"The variation in mineralogy with particle size leads to variations in $n_s$."*

*P15 L29-32: Awkward sentence. Rephrase.*

Sentence rephrased (p.16, l.33-p.17, l.1 of the revised manuscript):

"The focus of earlier IN studies on the clay mineral IN behavior is supported by the observation that three of the four airborne samples (Crete, Peloponnese and Tenerife), which had been long-range transported and were almost entirely in the size range < 2.5 µm, had the highest clay mineral fraction."

"*Three of the four airborne samples (Crete, Peloponnese and Tenerife) had the highest clay mineral content and were amongst the least ice nucleation-active samples.*"

*P16 L 12: "can not" ➔ "cannot" (to be consistent with the rest of the manuscript)*

done

*P16 L 21-22: Redundant (P4 L22)*

We deleted the sentence.

**References:**

Boose, Y.  et al.: Heterogeneous ice nucleation on dust particles sourced from 9 deserts worldwide - Part 2: Deposition and condensation freezing, in prep., 2016

Hiranuma, N., Augustin-Bauditz, S., Bingemer, H., Budke, C., Curtius, J., Danielczok, A., Diehl, K., Dreischmeier, K., Ebert, M., Frank, F., Hoffmann, N., Kandler, K., Kiselev, A., Koop, T., Leisner, T., Möhler, O., Nillius, B., Peckhaus, A., Rose, D., Weinbruch, S., Wex, H., Boose, Y., DeMott, P. J., Hader, J. D., Hill, T. C. J., Kanji, Z. A., Kulkarni, G., Levin, E. J. T., McCluskey, C. S., Murakami, M., Murray, B. J., Niedermeier, D., Petters, M. D., O'Sullivan, D., Saito, A., Schill, G. P., Tajiri, T., Tolbert, M. A., Welti, A., Whale, T. F., Wright, T. P., and Yamashita, K.: A comprehensive laboratory study on the immersion freezing behavior of illite NX particles: a comparison of 17 ice nucleation measurement techniques, Atmos. Chem. Phys., 15, 2489–2518, doi:10.5194/acp-15-2489-2015, 2015.

Hoyle, C. R., Pinti, V., Welti, A., Zobrist, B., Marcolli, C., Luo, B., Hoskuldsson,  A., Mattsson, H. B., Stetzer, O., Thorsteinsson, T., Larsen, G., and Peter, T.: Ice nucleation properties of volcanic ash from Eyjafjallajokull, Atmos. Chem. Phys., 11, 9911–9926, doi:10.5194/acp-11-9911-2011, 2011.

Ickes, L., Welti, A., Hoose, C., and Lohmann, U.: Classical nucleation theory of homogeneous freezing of water: thermodynamic and kinetic parameters, Phys. Chem. Chem. Phys., 17, 5514– 5537, doi:10.1039/C4CP04184D, 2015.

Kaufmann, L., Marcolli, C., Hofer, J., Pinti, V., Hoyle, C. R., and Peter, T.: Ice nucleation efficiency of natural dust samples in the immersion mode, Atmos. Chem. Phys., 16, 11177-11206, doi:10.5194/acp-16-11177-2016, 2016

Kumai, M.: Snow crystals and the identification of the nuclei in the Northern United States of America, J. Meteor., 18, 139-150, doi: 10.1175/1520-0469(1961)018<0139:SCATIO>2.0.CO;2, 1961.

Zhu, C., Veblen, D.R., Blum, A. E., Chipera, S. J.: Naturally weathered feldspar surfaces in the Navajo Sandstone aquifer, Black Mesa, Arizona: Electron microscopic characterization, Geochim. Cosmochim. Acta, 70, 18, 4600-4616, doi: 10.1016/j.gca.2006.07.013, 2006.

---

## Author Response (AR2)

We thank Reviewer 2 and the Editor for their additional comments. We indicate the reviewer's remarks in blue in the following. Amended versions of the paper are given in *italics* for new sections and smaller red text for the original text.

1) The last sentence of the first paragraph in Sect. 4 (Conclusions) is not intuitive… "We 'cannot' therefore conclude that microcline is generally atmospherically 'not' relevant…" – do you imply that microcline is relevant?

We have rephrased the sentence (p.16, l.17-19):

"*A conclusion on the atmospheric relevancy of microcline is therefore not possible because even in low amounts - of a few percent - it could nucleate ice and glaciate clouds at temperatures warmer than 253 K.*"

We hope this clarifies that based on our method a general conclusion on the role of microcline vs. orthoclase for atmospheric ice nucleation is not possible.

2) Given large uncertainties in the quartz IN as presented in Fig. 5 as well as limitations provided in Sect. 3.3 (last paragraph), referee #2 is still not convinced for the importance of quartz on overall dust IN. The 'correlation' shown in Table 5 and supplemental figures seems qualitative (not quantitative… could be a good coincidence).

We believe we have discussed the caveats of the correlation method of bulk mineralogy and ice nucleation activity sufficiently in the manuscript (p.14, l.27 and p.15, l.5-9). Despite these caveats we think the method is useful in that the correlations indicate trends, namely that with increasing quartz content the ice nucleation activity of a dust sample increases at temperatures below 250 K. The correlations with quartz and with the sum of quartz and feldspar are significant at the 0.05 level at 238 and 240 K even when only the six samples are included for which we know that the mineralogy is fully representative for particles in the size range we investigated for ice nucleation. Of course, correlation is not necessarily causation and the observed trends may be coincidence. But this is true for any correlation method. Further, we think that we have interpreted our results in a cautious manner, e.g. p. 14, l.7: "Nevertheless, the overview of the correlation coefficients gives *an idea* of the effect of certain minerals on $n_s$."

What else could explain for example the high ice nucleation ability of the Australia sample vs. that of the Atacama milled sample? The Australia sample contains about 8 % K-feldspar and 91 % quartz. In contrast the Atacama milled sample contains 22 % K-feldspar, 43 % Na-plagioclase and 10 % quartz. Both samples were milled before the ice nucleation experiments and for both samples an influence of organic material was excluded by heat treatment. The Australia sample shows almost an order of magnitude higher $n_s$ at any temperature between 241 and 246 K. If quartz played no role for the ice nucleation ability only the K-feldspar would be left to explain the high $n_s$. This would mean that in the case of the Australia sample, the 8 % K-feldspar must be distributed over more than 90 % of the particles (based on the frozen fraction results in Fig. 4a of the manuscript), while the 22 % K-feldspar and the 43 % Na-plagioclase were either not internally mixed with other minerals (hence only present in maximum 65 % of the particles) or covered by other, less active minerals. This very different behaviour of the feldspar in the two samples seems rather unlikely. It appears much more realistic, that the quartz content contributes to the ice nucleation activity as shown for example by Atkinson et al. 2013. Their Figure 1 is reproduced here and the temperature interval between 241 and 246 K is marked with a red box. It is apparent that in this temperature range quartz becomes similarly ice-active as the feldspars supporting our findings for natural dust samples.

Our empirical ice nucleation results show that quartz contributes to the ice nucleation activity of the natural dust samples at these comparably low temperatures. Instead of discussing the role of each mineral for the $n_s$ of each dust, the correlation analysis is used to find overall trends. How exactly quartz leads to ice nucleation and why different laboratory studies show different results for the ice nucleation activity of quartz, are topics for future studies.

[Figure]

Figure 1: Experimental freezing results for individual minerals (Atkinson et al. 2013).

**Further Remarks:**

As part of further investigations on additional airborne collected samples from the Izaña observatory, Tenerife, we found indications that they contained a significant amount of smectite. Based on this finding we reanalyzed all dust samples from this manuscript again with respect to smectite and performed a glycol treatment test for those samples where indications for smectite were found. The glycol treatment test unambiguously shows the presence of smectite. For samples where the test confirmed its presence, smectite was then included in the Rietveld fit. This showed that the Mojave and Tenerife samples contained 26.1 wt% and 31.8 wt% smectite, respectively. Inclusion of smectite in the Rietveld fit resulted mostly in a decrease in relative abundance of illite (-9.4 wt% and -8 wt%) in the Tenerife and Mojave sample respectively, kaolinite (-8.7 wt%) in the Tenerife sample  and Na-plagioclase (-6.3 wt%) in the Mojave sample compared to the original analysis not including smectite.

We recalculated the correlation coefficients for Tables 5 and 6. In Table 5, $R$ changed at most by 0.07 but in most cases only by ≤ 0.03. In Table 6 the changes were larger (up to 0.37) due to the lower number of samples included

in the correlations. However, these relatively large changes occurred for $R$ values which were negative or very close to 0. For the high positive $R$´s, the change in $R$ was ≤ 0.16. The inclusion of smectite in the Rietveld fit did not change the observed trends that quartz and feldspars are the best predictors for $n_s$.

Accordingly, we have rephrased the respective sentences:

p.13, l.13-15: "The Saharan samples show a great variety with the Tenerife sample having the highest content of clay minerals (illite + kaolinite + palygorskite: 43.4 wt%) of all dusts, similar to the findings of Alastuey et al. (2005). "

Now p.13, l.13-15: "*The Saharan samples show a great variety with the Tenerife sample having the highest content of clay minerals (illite + kaolinite + smectite + palygorskite: 57.7 wt%) of all dusts, similar to the findings of Alastuey et al. (2005).*

p.16,l.14-15: "Microcline was found in one airborne sample (6 wt%) and in surface-collected samples from different locations."

Now p.16, l.14-15: "*Microcline was found in one airborne sample (4 wt%) and in surface-collected samples from different locations.*"

p.14,l.34-35: " The correlation with illite and kaolinite is still negative at any temperature despite the fact that some of the samples in this subset contain a comparably large amount of clay minerals (Tenerife: 38.5 wt%, Peloponnese: 20.2 wt%)."

Now p.14,l.34-35: " *The correlation with illite and kaolinite is still negative at any temperature despite the fact that some of the samples in this subset contain a comparably large amount of these clay minerals (Tenerife: 21.8 wt%, Peloponnese: 20.2 wt%).*"

Furthermore, we have replotted Figure 1 in the SOM.

We also corrected the GPS coordinates of the Etosha sample in the SOM, after they had been corrected in the final version of Kaufmann et al. 2016.

**References:**

[revised manuscript text omitted]